# PI3K signaling specifies proximal-distal fate by driving a developmental gene regulatory network in SOX9+ mouse lung progenitors

**Divya Khattar[1†], Sharlene Fernandes[2,3†], John Snowball[2,3], Minzhe Guo[2,3], Matthew C Gillen[1], Suchi Singh Jain[2,4], Debora Sinner[2,3,5], William Zacharias[2,3,6], Daniel T Swarr[1,2,3,5]***

[1]Department of Pediatrics, University of Cincinnati, Cincinnati, United States; [2]Perinatal Institute, Cincinnati Children's Hospital Medical Center, Cincinnati, United States; [3]Division of Neonatology, Perinatal and Pulmonary Biology, Cincinnati Children's Hospital Medical Center, Winston-Salem, United States; [4]Wake Forest University, Winston-Salem, United States; [5]Department of Pediatrics, University of Cincinnati, Cincinnati, United States; [6]Department of Medicine, University of Cincinnati, Cincinnati, United States

**\*For correspondence:**
Daniel.Swarr@cchmc.org

[†]These authors contributed equally to this work

**Competing interest:** The authors declare that no competing interests exist.

**Abstract** The tips of the developing respiratory buds are home to important progenitor cells marked by the expression of SOX9 and ID2. Early in embryonic development (prior to E13.5), SOX9+progenitors are multipotent, generating both airway and alveolar epithelium, but are selective progenitors of alveolar epithelial cells later in development. Transcription factors, including *Sox9, Etv5, Irx, Mycn,* and *Foxp1/2* interact in complex gene regulatory networks to control proliferation and differentiation of SOX9+progenitors. Molecular mechanisms by which these transcription factors and other signaling pathways control chromatin state to establish and maintain cell-type identity are not well-defined. Herein, we analyze paired gene expression (RNA-Seq) and chromatin accessibility (ATAC-Seq) data from SOX9+ epithelial progenitor cells (EPCs) during embryonic development in *Mus musculus*. Widespread changes in chromatin accessibility were observed between E11.5 and E16.5, particularly at distal cis-regulatory elements (e.g. enhancers). Gene regulatory network (GRN) inference identified a common SOX9+ progenitor GRN, implicating phosphoinositide 3-kinase (PI3K) signaling in the developmental regulation of SOX9+ progenitor cells. Consistent with this model, conditional ablation of PI3K signaling in the developing lung epithelium in mouse resulted in an expansion of the SOX9+ EPC population and impaired airway epithelial cell differentiation. These data demonstrate that PI3K signaling is required for epithelial patterning during lung organogenesis, and emphasize the combinatorial power of paired RNA and ATAC seq in defining regulatory networks in development.

## Editor's evaluation

This study reveals the importance of PI3K in regulating early aspects of lung development including the establishment of a proximal-distal gradient in cell fate in the lung endoderm. The data presented will provide a rich resource for further examination of the role of PI3K and the transcription factor *Sox9* in lung endoderm development.

**eLife digest** Studying how lungs develop has helped us understand and treat often-devastating lung diseases. This includes diseases like cystic fibrosis which result from spelling mistakes known as mutations in a person's genetic code. However, not all lung diseases involve mutations. Many other diseases, in both adults and children, are caused by genes failing to switch on or off at some point during lung development.

DNA is surrounded by various proteins which package it into a compressed structure known as chromatin. Cells can control which genes are turned on or off by modifying how tightly packed parts of the genetic code are within chromatin. Changes in chromatin accessibility, also known as 'epigenetic' changes, are a normal part of development, and guide cells towards specific jobs or identities as an organ matures. However, how this happens in the developing lung is poorly understood.

Here, Khattar, Fernandes et al. set out to determine how chromatin accessibility shapes development of the tissue lining the lungs, focusing on a group of progenitor cells which produce the protein SOX9. These cells are initially found at the tips of the early lung, where they go on to develop into the cells that line the whole of the mature organ.

Initial experiments used large-scale genetic techniques to measure gene activity and chromatin accessibility simultaneously in progenitor cells extracted from the lungs of mice. Khattar, Fernandes et al. were then able to predict the signaling pathways that shape the lung lining based on which genes were surrounded by unpacked chromatin, and determine the proteins responsible for these epigenetic changes. This included the signaling pathway Phosphatidylinositol 3 kinase (PI3K) which is involved in a number of cellular processes.

Additional experiments in mice confirmed that the PI3K pathway became active very early in lung development and remained so until adulthood. In contrast, mice lacking a gene that codes for a key part of the PI3K pathway had defective lungs which failed to develop a proper lining.

The data generated in this study will provide an important resource for future studies investigating how epigenetic changes drive normal lung development. Khattar, Fernandes et al. hope that this knowledge will help researchers to better understand the cause of human lung diseases, and identify already available 'epigenetic drugs' which could be repurposed to treat them.

## Introduction

Morphogenesis of the mammalian lung begins as an outpouching of a simple bud from the embryonic foregut, which branches and extends into the splanchnic mesoderm as the primitive respiratory tubules are formed. The tips of the lung bud are lined by highly proliferative epithelial progenitor cells marked by expression of the genes *Sox9* and *Id2* (**Rawlins et al., 2009**; **Morrisey and Hogan, 2010**). Over the course of early lung development, SOX9$^+$ progenitor cells give rise to the entire complement of lung epithelium, from proximal airway epithelial cells to distal type I (AT1) and type II alveolar (AT2) epithelial cells (**Rawlins et al., 2009**; **Laresgoiti et al., 2016**; **Nichane et al., 2017**; **Nikolić et al., 2017**; **Miller et al., 2018**). Remarkable progress has been made in understanding the signaling pathways (e.g. *Wnt, Bmp*, *Shh*, retinoic acid [RA], and FGF signaling) and transcription factors that control development, differentiation, and maturation of the lung epithelium (**Morrisey and Hogan, 2010**; **Swarr and Morrisey, 2015**; **Whitsett et al., 2019**). Present understanding of how these various signals are integrated to establish respiratory epithelial cell identity remains limited; how this 'cellular memory' is disrupted over the course of the human lifespan during homestasis, repair after injury, and in various lung diseases is an area of active investigation.

Chromatin state serves as a medium through which various developmental and external stimuli are integrated into comprehensive cellular programs which enable regulatory proteins access to activate or repress gene expression. Epigenetic programs controlling progenitor cell identies are maintained in spite of repeated rounds of cellular division, fluctuating transcription factor levels, or dynamic inputs from various signaling pathways, (**Klemm et al., 2019**; **Sheikh and Akhtar, 2019**). Chromatin accessibility changes dynamically throughout development, to establish and maintain cellular identity in a variety of cell types and organ systems (**Gorkin et al., 2017**; **Trevino et al., 2020**). The extent to which changes in chromatin accessibility contribute to lung development as a diversity of mature epithelial cells differentiate from embryonic progenitors remains incompletely understood.

Chromatin accessibility, or 'the degree to which nuclear macromolecules are able to physically interact with chromatinized DNA' can now be measured in a relatively small number of cells using Assay for Transposible-Accessible Chromatin using sequencing (ATAC-Seq) (*Buenrostro et al., 2013*; *Klemm et al., 2019*), wherein Tn5 transposase inserts Illumina sequencing adapters into accessible regions of chromatin. Although accessible chromatin regions only make up a minority of the genome (2–3%), they account for a high proportion of the genomic loci bound by transcription factors, and are highly correlated with cis-regulatory elements (CREs) identified by other methods, such as promoters, enhancers, insulators, and silencers (*Buenrostro et al., 2013*; *Buenrostro et al., 2015b*).

Herein, we used paired RNA-Seq and ATAC-Seq to define the chromatin accessibility landscape of developing SOX9+ lung epithelial progenitor cells at multiple development timepoints. Acessible chromatin regions in SOX9+ cells were highly correlated with lung-specific histone post-translational modifications (PTMs) found on validated cis-regulatory elements. Dynamic changes in these accessible chromatin regions were correlated with gene expression changes observed during the course of SOX9+ lung epithelial cell differeniation. Chromatin accessibility changes were observed in or near genes associated with phophatidylinositol 3-kinase (PI3K) signaling. Genetic ablation of PI3K signaling within the lung epithelium caused expansion of the SOX9+ progenitor cell population at the expense of mature lung epithelial cell differentiation. Our results demonstrate the power of epigenomic analysis to identify new roles for gene regulatory networks and signaling pathways in lung epithelial development, highlighting the previously unappreciated role of PI3K signaling in SOX9+ cell differentiation and proximal-distal cell fate specification.

## Results

### The multi-potent SOX9+ lung epithelial progenitor cell (EPC) population

Shortly after lung specification occurs, highly proliferative distal progenitor cells marked by the expression of SOX9+ appear at the tips of the growing lung buds (*Figure 1A–I*). A suite of transcription factors (eg, *Id2*, *Etv5*, *Foxp1/2*, *Mycn*) play critical roles in the development of the SOX9+ distal tip progenitor cell population (*Figure 1D*; *Morrisey and Hogan, 2010*; *Swarr and Morrisey, 2015*). Several lines of evidence, including lineage tracing and 3D culture model systems, demonstrate that early in development SOX9+ progenitor cells are multi-potent, generating both airway and distal alveolar epithelium (*Rawlins et al., 2009*; *Laresgoiti et al., 2016*; *Nichane et al., 2017*; *Nikolić et al., 2017*). In order to orthogonally validate these observations and gain insight into the transcriptional changes occurring during these developmental decisions, we peformed pseudotime analysis (Monocle) of RNA-Seq data from mouse lung epithelium from E11.5 to E16.5 (*Trapnell et al., 2019*). Starting from E11.5, two distinct pseudotime branches arise from SOX9+ progenitors, terminating in airway and distal lung epithelial cells at E16.5 (*Figure 1J–K*, *Figure 1—figure supplement 1A, B*). We used Monocle to find modules consisting of co-regulated genes during distal differentiation, revealing 26 unique gene modules. A subset of these gene modules overlap with pseudotime analysis and correspond with specific aspects of SOX9+ epithelial differentiation (*Figure 1L–M*, and *Figure 1—figure supplement 1*). One such gene module consists of genes highly expressed E11.5 lung and preserved the distal differential branch of the Monocle pseudotime analysis; these genes are involved in lung epithelial development and early branching, including *Sox9*, *Wnt7b*, *Bmp4*, and *Hmga2*, validating the significance of both the pseudotime analyses and gene module predictions while also identifying other coordinately expressed genes that may be involved in EPC differentiation (module 8, *Figure 1L–M*). Other modules highlight genes that likely regulate the proliferating progenitor cells on the same central branch to distal epithelium (module 16, *Figure 1—figure supplement 1E, G*), and genes involved in the function of differentiated airway epitheilum (modules 4 and 12, *Figure 1—figure supplement 1C, D, F*). These data are consistent with gene set enrichment analysis of bulk RNA-Seq data at E11.5 and E16.5 (*Figure 2—figure supplement 1*).

In order to evaluate the landscape and dynamics of cis-regulatory DNA elements associated with these transcriptional changes occurring during development of EPCs, SOX9+ progenitor cells (SOX9-GFP$^+$, CD326$^+$, CD31$^-$, CD45$^-$, 7-AAD$^-$) were isolated by fluorescence-activated cell sorting (FACS) from a transgenic reporter mouse (containing an EGFP allele driven by the Sox9 promoter element) (*Gong et al., 2017*; *Figure 2—figure supplement 2*). SOX9+ progenitors were then

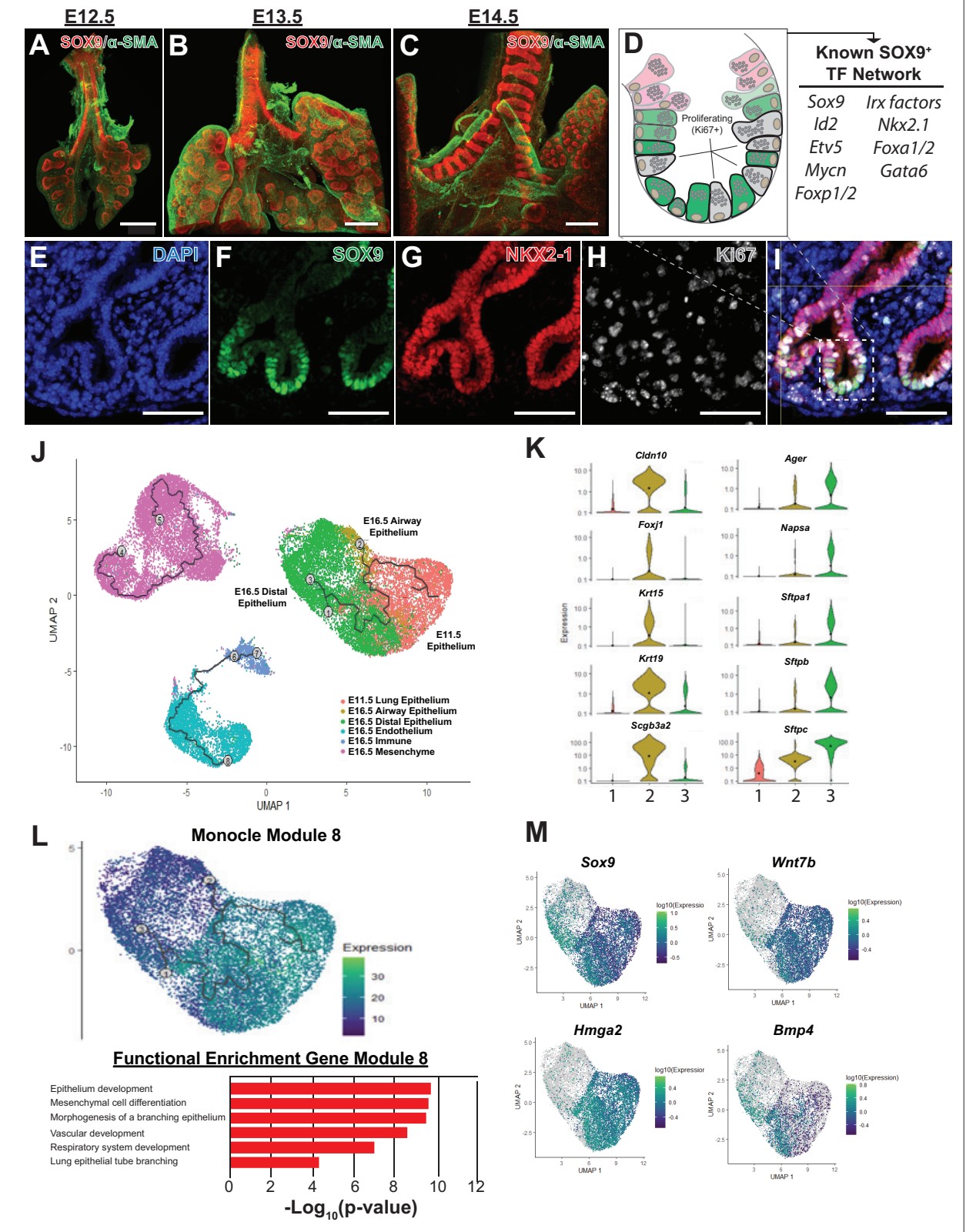

**Figure 1.** The SOX9+multi-potent lung epithelial progenitor cell population. (**A–C**) Whole mount confocal imaging of embryonic lungs isolated from SOX9-GFP reporter mice at E12.5 (**A**), E13.5 (**B**), and E14.5 (**C**) show SOX9 expression at the distal branch tips and in the tracheal mesenchyme. (**D**) Schematic summarizing the transcription factors known to play an important role in the development and differentiation of the SOX9+progenitors. (**E–I**) Immunofluoresence microscopy shows high levels of SOX9 staining in the epithelium of the distal lung tips, with a high percentage of

*Figure 1 continued on next page*

Figure 1 continued

Ki67 +proliferating cells early in development. (**J–M**) Monocle3 lineage analysis was performed on E11.5 FACS-sorted lung epithelium and E16.5 whole lung. (**J**) Lineage analyses predict E11.5 lung epithelial cells give rise to both proximal and distal lung epithelial cells at E16.5. (**K**) Known distal and proximal genes markers are expressed in their corresponding cell clusters at E16.5 but not E11.5, validating cluster identification. Cluster labels: 1 – E11.5 lung epithelium; 2 – E16.5 airway epithelium; 3 – E16.5 distal epithelium. (**L**) Monocle 3 analysis of isolated epithelial populations identifies multiple gene modules with specific expression patterns. One module of gene expression is consistent with the predicted distal differentiation trajectory and consists of genes related to processes important to lung development, including epithelial/respiratory system branching and development. (**M**) The expression patterns of example genes in this module are visualized and include genes known to be essential for proper distal lung development including *Sox9*, *Wnt7b*, *Hmga2,* and *Bmp4*. Scale bars: 200 μm (**A–C**), 50 μm (**E–I**).

The online version of this article includes the following figure supplement(s) for figure 1:

**Figure supplement 1.** Single-cell analysis of mouse embryonic lung.

subjected to bulk RNA-Seq and ATAC-Seq. E11.5 and E16.5 were chosen to capture cells prior to, and shortly after, the period of proposed restriction from multi-potent progenitor to distal alveolar progenitor cells (*Rawlins et al., 2009*; *Nichane et al., 2017*; *Nikolić et al., 2017*; *Frank et al., 2017*; *Figure 2A*, *Supplementary file 1*).

## Accessible regions of chromatin in SOX9+ EPCs are active cis-regulatory elements that correlate with neighboring gene expression

Analysis of chromatin accessibility by ATAC-Seq identified over 30,000 regions of open chromatin at each timepoint (E11.5, n=34,602; E16.5, n=40,823; *Figure 2B*); only peaks replicated in independent biological samples were included for further analysis (*Figure 2—figure supplement 3A, B*, *Supplementary file 2*; *Supplementary file 3*). The genome-wide distribution of ATAC-Seq peaks was consistent with previously published chromatin accessibility studies, showing a high-degree of enrichment near gene transcriptional start sites (TSS) (*Figure 2—figure supplement 3C-H*). The presence of ATAC-Seq peaks positively correlated with expression the nearest gene (RNA-Seq), and this correlation was strongest for peaks within promoter regions. Moreover, peaks only detected at a single developmental timepoint were associated with higher levels of expression of the nearst gene at that same developmental stage (for example, genes nearest to peaks unique to the E11.5 developmental stage overall were observed to have higher expression levels at E11.5 compared to E16.5; *Figure 2E–F*, *Figure 2—figure supplement 4A-E*).

To determine whether these accessible regions of chromatin had shared features of active cis-regulatory DNA elements (CREs), we examined H3K4me$^3$, H3K4me$^1$, and H3K27ac ENCODE ChIP-Seq read density (embryonic mouse, whole lung) across all identified ATAC-Seq peaks (*ENCODE Project Consortium, 2012*; *Davis et al., 2018*). Significant enrichment was observed for H3K4me$^3$ read density (cluster 1, as identified through unbiased hierarchial clustering) across peaks predicted to reside within gene promoter regions (*Figure 2C–D* and *Figure 2—figure supplement 4F, G*). A second subset of ATAC-Seq peaks (cluster 2) were enriched for either both H3K4me$^1$ and H3K27ac read density, or H3K4me$^1$ ready density alone, consistent with an 'active' or 'decommissioned' enhancer states, respectively (*Jadhav et al., 2019*). (*Figure 2C–D* and *Figure 2—figure supplement 4F, G*) Taken together, these data suggest that regions identified by ATAC-Seq represent a bona fide map of active cis-regulatory elements (e.g. promoters, active and decommissioned enhancers) within the SOX9+ lung epithelial progenitor cells, and that these regulatory elements are regulating the transcriptional dynamics observed during EPC development.

## Developmental changes in SOX9+ EPC chromatin accessibility preferentially occur outside of promoter regions

Nearly a quarter of accessible chromatin regions were unique to each developmental timepoint, a remarkable observation given that the two cell populations are SOX9+ lung epithelial progenitor cells separated by just 5 days of development (*Figure 2A*). We used the MACS2 bdgdiff tool to identify regions of chromatin that were more accessible at E11.5 (n=18,582), more accessible at E16.5 (n=11,980), or did not undergo significant changes in accessibility between the two timepoints (n=35,312) (hereafter referred to as E11.5-differentially accessible regions (DARs), E16.5-DARs, and common regions, respectively). (*Supplementary file 4*; *Supplementary file 5*; *Supplementary file 6*) As was observed for all peaks, both sets of DARs and common regions were significantly enriched

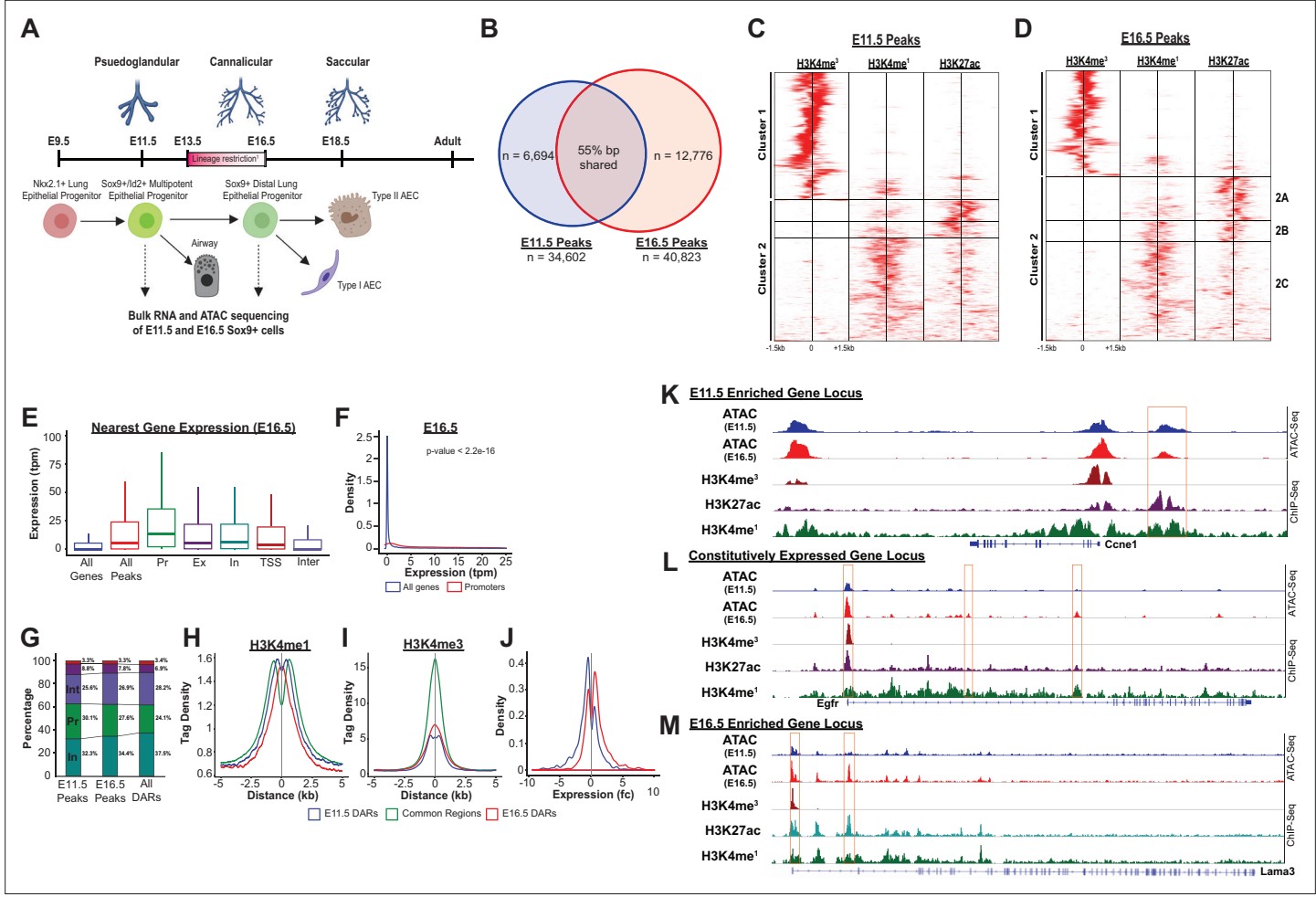

**Figure 2.** The SOX9+progenitor cell chromatin accessibility landscape. (**A**) Schematic strategy for isolation of SOX9+EPC. Cells sorted at E11.5 and E16.5 were used for RNA-seq and ATAC-seq analysis (n=2 biological replicates pooled from 8 to 12 embryos at each timepoint for ATAC-Seq, n=3 pooled biological replicates for RNA-Seq), which correspond to pseudoglandular and late saccular stages of lung development. (**B**) More than 34,000 regions of open chromatin were identified by ATAC-Seq at each developmental timepoint. (**C–D**) ChIP-Seq tag densities for H3K4me[3], H3K4me[1], and H3K27ac were plotted and clustered using Homer and R, respsectively. H3K4me[3] tag density corresponds to the accessible promoter regions (Cluster 1), and H3K4me[1] with or without H3K27ac correspond to active and 'decommissioned' enhancer regions (Clusters 2 A-C), respectively. (**E–F**) Gene expression levels correlate with the presence of a nearby region of open chromatin, particularly when it is located within the promoter region. Pr – Promoter; Ex – Exonic; In – Intronic; TSS – Transcriptional Stop Site; Inter – Intergenic (**G**) Differentially accessible chromatin regions (DARs) were more likely to be located with intronic (In) and intergenic (Int) regions instead of promoters. (**H**) H3K4me[1] peak enrichment is found at the center of DARs. (**I**) Common regions were more enriched for H3K4me[3] signal compared to DARs at either timepoint. (**J**) E11.5 and E16.5 DARs correlate with expression of nearby genes by RNA-seq. (**K–L**) Examples of three categories of genomic loci are shown: (**K**) Decreasing levels of accessibility over time for progenitor genes, (**L**) Changing accessibility without change in gene expression for housekeeping genes, and (**M**) Increasing accessibility at later developmental timepoints for genes required for function fo the mature lung.

The online version of this article includes the following figure supplement(s) for figure 2:

**Figure supplement 1.** Bulk RNA-Seq analysis of the developing SOX9+EPC population.

**Figure supplement 2.** FACS sorting strategy for isolation of Sox9+EPC cells.

**Figure supplement 3.** ATAC-Seq peak characteristics.

**Figure supplement 4.** Accessible chromatin regions, associated histone marks, and neighboring gene expression.

within 2 kb of transcriptional start sites (*Figure 2—figure supplement 4*). However, compared to peaks that did not undergo changes in chromatin accessibility, DARs were significantly more likely to be found within intronic and intergenic regions, and were less likely to be found within promoters (*Figure 2G* and *Figure 2—figure supplement 4I*). DARs at both E11.5 and E16.5 exhibited peak H3K4me[1] read density in close proximity to the center of the DAR. In contrast, there was relative

de-enrichment of H3K4me[1] read density at the center of common regions (and peaks flanking the common region by ~500 bp on each side) (*Figure 2H*). Similarly, read density for the promoter mark H3K4me[3] was significantly enriched and centered on common region peaks, compared to DAR peaks (*Figure 2I*). At the genome-wide level, changes in chromatin accessibility were significantly correlated with differential gene expression. Genes adjacent to regions of chromatin more accessible at E11.5 (E11.5-DARs) were significantly more likely to have higher mRNA expression at E11.5 compared to E16.5, and vice versa (*Figure 2J*). Examples of differentially accessible chromatin regions flanking or within two highly differentially expressed genes (e.g. Ccne1, Lama3), and common accessible regions within a stably expressed gene locus (e.g. Egfr), are shown in *Figure 2K–M*. These data support the concept that regions of chromatin accessibility identified in this study contain active CREs that influence neighboring gene expression. Moreover, although chromatin remodeling occurs across all types of cis-regulatory elements during development of SOX9+lung epithelial progenitors, developmental changes in chromatin accessibility were preferentially located outside of the promoter regions at distal cis-regulatory elements (e.g. enhancers). Collectively, these data also emphasize that chromatin accessibility at distal CREs such as enchancers likely plays a key and incompletely explored role in directing lung epithelial development.

## Chromatin remodeling is enriched near loci associated with PI3K signaling

Although analysis of gene expression alone enables predictions regarding signaling pathways and transcription factor networks driving transcriptional changes, we hypothesized that adding an understanding the cis-regulatory landscape within developing SOX9+ lung epithelial cells would provide additional mechanistic insights into how these trans-activating factors promote and maintain cell differentiation (*Klemm et al., 2019*; *Trevino et al., 2020*). To this end, we used the Paired Expression and Chromatin Accessibility (PECA) computational model to build a gene regulatory network for the SOX9+ EPC population at E11.5 and E16.5 (*Duren et al., 2017*). At the epigenomic level, this model compares open regions of chromatin to a map of known cis-regulatory elements identified through large-scale epigenomics consortia (e.g. ENCODE), TF motifs detected within these regions, and TF expression data derived from expression data to eliminate spurious motif elements (for example, motifs for transcription factors that are not expressed at the mRNA level in the paired mRNA-seq dataset). The model then compares these open elements to paired mRNA-seq data in order to make predictions about active TFs, the genes they regulate, the cis-regulatory elements that they bind to, and the chromatin regulatory complexes that they interact with (*Figure 3A*). The interactions between TFs predicted to regulate more than 100 target genes are shown in *Figure 3B*. Transcription factors with the highest degree of interconnectivity (top 25 TFs) are shown in *Figure 3C*. Well-studied transcription factors, including those highlighted in *Figure 1D*, such as *Id2*, *Irx* and *Foxp factors*, *Etv5*, and *Foxa2* appear as central nodes. Additional TFs play important roles in transcription generally (e.g. *E2f* family TFs, *Jun*). However, multiple other TFs that appear in this central network have been less-well studied in the contex of lung epithelial development. Intriguingly, several of these key nodes have previously shown to interface with the PI3K signaling pathway (*Figure 3D–G*). The transcription factor *Sox4*, which has been implicated in the pathogenesis of non-small cell lung cancer (NSCLC) and was recently found to be enriched in a regeneration associated pre-alveolar type-1 cell transitional state (PATS), has been shown to both regulate and have its own expression regulated by PI3K signaling (*Ramezani-Rad et al., 2013*; *Wang et al., 2019*; *Bilir et al., 2016*; *Mehta et al., 2017*; *Kobayashi et al., 2020*). Another central node in this network, *Hbp1,* is a known repressor of beta-catenin transactivation, but is also negatively regulated by PI3K signaling (*Coomans de Brachène et al., 2014*; *Bollaert et al., 2018*). Finally, *Grhl2*, a third central node in this network, was predicted to be regulated via the PI3K signaling component *Pten*. Previous work has shown *Ghrl2* promotes airway differentiation in the lung, and that loss of *Grhl2* in the lung leads to expansion of the SOX9+ EPC population (*Kersbergen et al., 2018*).

To complement the gene-regulatory network model predicted by PECA, we performed gene set enrichment analysis (GSEA) for both RNA-Seq and ATAC-Seq datasets, using the nearest neighbor gene-peak pair for ATAC data. KEGG pathway analysis of RNA-Seq data identified biological processes and pathways consistent with known regulators of lung epithelial development. At E11.5, gene categories involving cell proliferation (e.g. 'cell cycle', 'homologous recombination'), protein synthesis

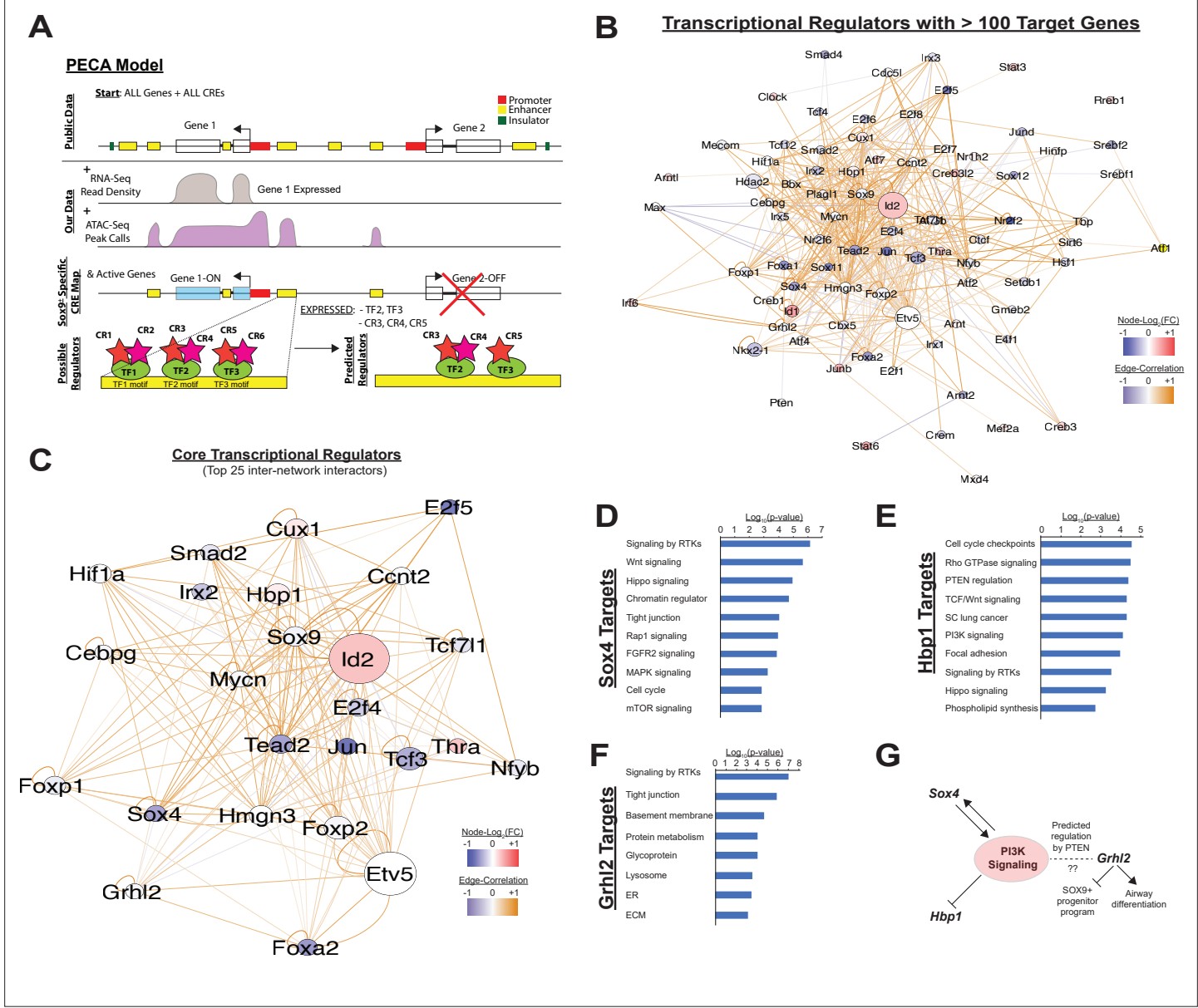

**Figure 3.** Paired expression and acessibility modeling of SOX9+progenitor cells. (**A**) The computational model Paired Expression and Chromatin Accessibility (PECA) was used to develop a SOX9+ epithelial progenitor cell gene regulatory network. Potential active cis-regulatory elements, key transcription factor (TF) networks, and chromain regulatory (CR) protein complexes active in the SOX9+ EPCs cells are predicted. (**B**) The co-regulatory relationships between transcription factors with more than 100 predicted target genes are shown. Network nodes (TFs) are color coded according to their expression log2 fold-change between E11.5 to E16.5 (blue – decreasing expression, red – increasing expression), and network edges are color coded according to the correlation coefficient between the two indicated TFs (blue and organge indicate negative and positive regulatory relationships, respectively). (**C**) The transcription factors with the top 25 highest degree of network interactions in EPCs. (**D–E**) The TFs *Hbp1* and *Sox4* interact with PI3K signaling. Gene ontology analysis of predicted targets of these TFs in the SOX9+ progenitor cell network are shown. (**F**) *Grhl2* has previously been shown to promote airway differentiation, and its loss leads to expansion of the SOX9+ progenitor population. Our network data predict regulation of *Grhl2* expression through the PI3K signaling pathway. (**G**) Putative relationships between the PI3K signaling pathway and several key nodes of the SOX9+ progenitor cell gene regulatory network.

(e.g. 'ribosome', 'biogenesis'), and 'signaling pathways regulating pluripotency' are most enriched. By E16.5, processes related to development of functions associated with the mature distal lung epithelium, such as surfactant production (e.g. 'phagosome', 'lysosome', 'sphingolipid metabolism', 'protein export'), epithelial barrier function (e.g. 'ECM-receptor interaction', 'focal adhesion', 'collective duct

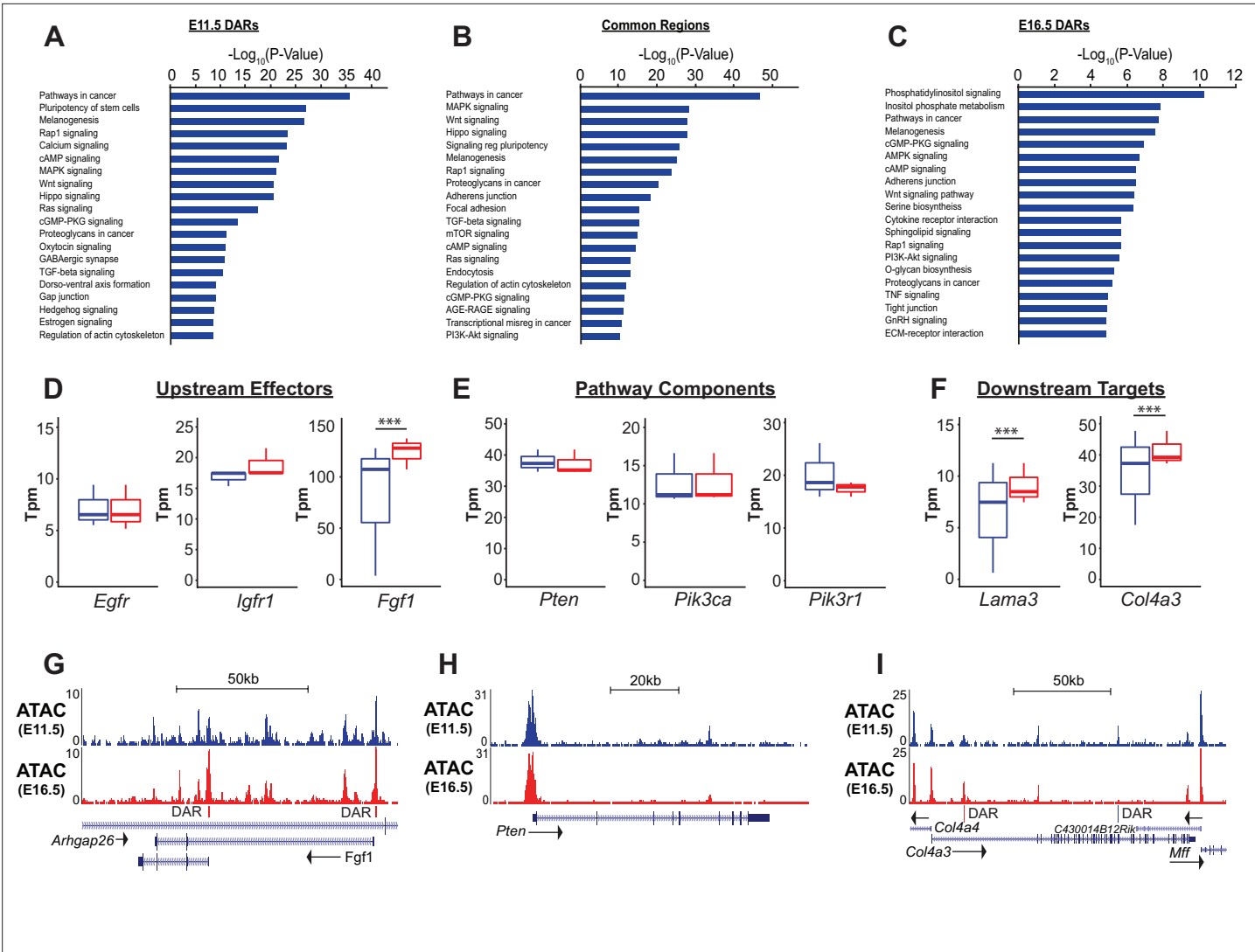

**Figure 4.** PI3K signaling in ATAC data. (**A–C**) Gene ontology analysis for genes nearest E11.5 and E16.5 differentially accessible regions (DARs) and common regions are shown. Pathways regulating lung epithelial development, such as Wnt, Hippo, and Hedgehog signaling are identified. A number of GO catatories associated with PI3K signaling were over-represented in genes adjacent to E16.5 DARs, as well as those nearby common regions. (**D–I**) Examples include proteins upstream of PI3K signaling (**D,G**), core pathway components (**E,H**), and known downstream targets of PI3K sigaling (**F,I**). Many of the downstream targets are components of epithelial basement membrane, which increase as development proceeds. (**G–I**) Examples of DARs correlated with gene expression changes for *Fgfr1* and *Col4a3* are shown. *Pten*, which does not change in expression between E11.5 and E16.5 has common accessible regions of chromatin (in promoter and 5ᵗʰ intron). E11.5 is shown in blue, E16.5 in red. Two-tailed Student's t-test used for D-F. *** p<0.001.

secretion'), and a switch in energy metabolism (e.g. 'oxidative phosphorylation', 'citrate/TCA cycle') are apparent (**Figure 2—figure supplement 1**).

While similar biological processes were enriched in both the ATAC-Seq and RNA-Seq data, analysis of ATAC-Seq data identified activation of a number of signaling pathways that were less apparent by RNA analysis alone. Signaling pathways known to be important in lung epithelial development, such as Wnt, *Shh*, Notch, and Hippo signaling were identified in both analyses. However, analysis ATAC-Seq data identified a potential role for a number of additional pathways in the development and differentiation of the SOX9+ lung epithelial progenitor cells, including cAMP, *Rap1*, *Ras*, MAPK, TGF-beta, and, importantly, PI3K-mTOR signaling (**Figure 4A–C** and **Figure 2—figure supplement 1B, C**).

Consistent with both the PECA and GSEA analyses, genes nearest to increasingly accessible regions of chromatin (E16.5-DARs) as EPC maturation proceeded were most strongly enriched in genes associated with phosphatidylinositol (KEGG mmu04070) and PI3K signaling (KEGG mmu04151).

Genes nearest to common accessible regions were also significantly enriched in these same pathways (*Figure 4B*). Components of the PI3K signaling pathway (eg. *Pik3ca*, *Pikap1*, *Pik3r1*, *Pten*), upstream ligand/receptors (eg. *Egfr*, *Fgf1*, *Igf1r*) and known downstream targets of PI3K signaling were associated with common accessible regions and E16.5 DARs (*Figure 4D–F*). While expression of many of core pathway components did not change between E11.5 and E16.5, expression of a number of potential upstream receptor/ligands (e.g. *Igfr*, *Fgf1*) and downstream targets of PI3K signaling increased, including major components of the epithelial basement membrane (e.g. *Lama3*, *Col4a3*). For example, differentially accessible regions within two promoter regions of the *Fgf1* locus and a E16.5-DAR within the first intron of the *Col4a3* locus are shown in *Figure 4G, I*. In contrast, common accessible chromatin regions are seen within the promoter and 5th intron of *Pten*, which is stably expressed between E11.5 and E16.5 in the SOX9+lung epithelium (*Figure 4H*). Taken together, these data suggested that the dynamic changes in chromatin state observed during SOX9+EPC development were modulated by PI3K signaling, an observation only possible with combined chromatin and RNA analysis.

## Epithelial-specific deletion of the *Pik3ca* gene in vivo causes persistence of SOX9+ EPCs at E18.5 and inhibition of epithelial differentiation

In order to evaluate the hypothesis that PI3K pathway activation occurs differentially throughout the developing lung epithelial cell population, we measured PI3K pathway activity by detecting AKT protein phosphorylated at Ser-473 (pAKT) in the embryonic mouse lung at E12.5 and E18.5, and in the adult lung (*Alessi et al., 1996*; *Zhang et al., 2016*; *Figure 5*). The highest levels of pAKT expression, and PI3K pathway activation, were observed in the proximal lung epithelium as early as E12.5 (*Figure 5A–C*) and persisted in the airway epithelium through adulthood (*Figure 5J–L*). Interestingly, early in development (E12.5) pAKT was more broadly expressed throughout the lung epithelium, and there appear to be a 'bimodal' pattern of distribution, with high expression levels in the proximal epithelium and a second peak of signal intensity at the distal branch tips. Much lower levels of pAKT expression and PI3K activation were observed in the mesenchyme at all ages, and the distal lung epithelium at later stages of development (*Figure 5D–L*). These observations are consistent with a recent published study of early postnatal lung, which demonstrated highest levels of pAKT at postnatal day 5 within the airway epithelium (*Zhang et al., 2020*). These data support the concept that a proximal-to-distal gradient of PI3K signaling within the lung epithelium may be required for lung epithelial differentiation and proper establishment of epithelial cell-type identity.

To test the hypothesis that epithelial activation of PI3K signaling is required for normal differentiation and proximal-distal patterning of the lung epithelium, we used *Shh*-Cre x *Pik3ca*^f/f^ mice to conditionally delete the *Pik3ca* gene from the embryonic mouse lung epithelium (*Harfe et al., 2004*; *Zhao et al., 2006*). (*Figure 6—figure supplement 1*) Lungs from conditional knockout mice (hereafter referred to as *Pik3ca cKO*mice) were harvested at 2-day intervals from E12.5-E18.5, along with littermate controls (including mice both with and without the Cre allele). The lungs of *Pik3ca cKO* embryos at E18.5 were remarkably mispatterned, with abnormal cysts lined by NKX2.1+epithelial cells scattered throughout the lungs (*Figure 6*), with the remainder of the lungs filled with simplified alveolar structures lined by increased numbers of SOX9+ lung epithelial cells (*Figures 6 and 7*). A significant increase in the number of Sox9+ epithelial cells was only observed at E18.5; in fact, the mean proportion of epithelium expressing Sox9 at E12.5 and E14.5 was slightly lower in *Pik3ca cKO* embryos (although this did not reach statistical significance). No significant differences were observed in the number of proliferative epithelial cells. Taken together, these data suggest a persistence (rather than proliferative expansion) of the SOX9+epithelial progenitor cell population through the end of fetal lung development.

One of the most dramatic aspects of the phenotype observed in *Pik3ca cKO* lungs was the marked reduction in the number of differentiated secretory and ciliated airway epithelial cells. (*Figure 8*) There was a decrease in both the total number of SCGB1A1+ secretory cells, and the intensity of SCGB1A1 staining within individuals cells, as measured by immunofluorescence microscopy. (*Figure 8Y-MM, QQ*) Moreover, there was a dramatic reduction in the mRNA levels of the secretory cell markers *Scgb1a1* and *Scgb3a2*. (*Figure 8PP*) A similar reduction was observed in the total number of ciliated cells, altered expression patterns of the microtubule protein TUB1A1, and a marked reduction on

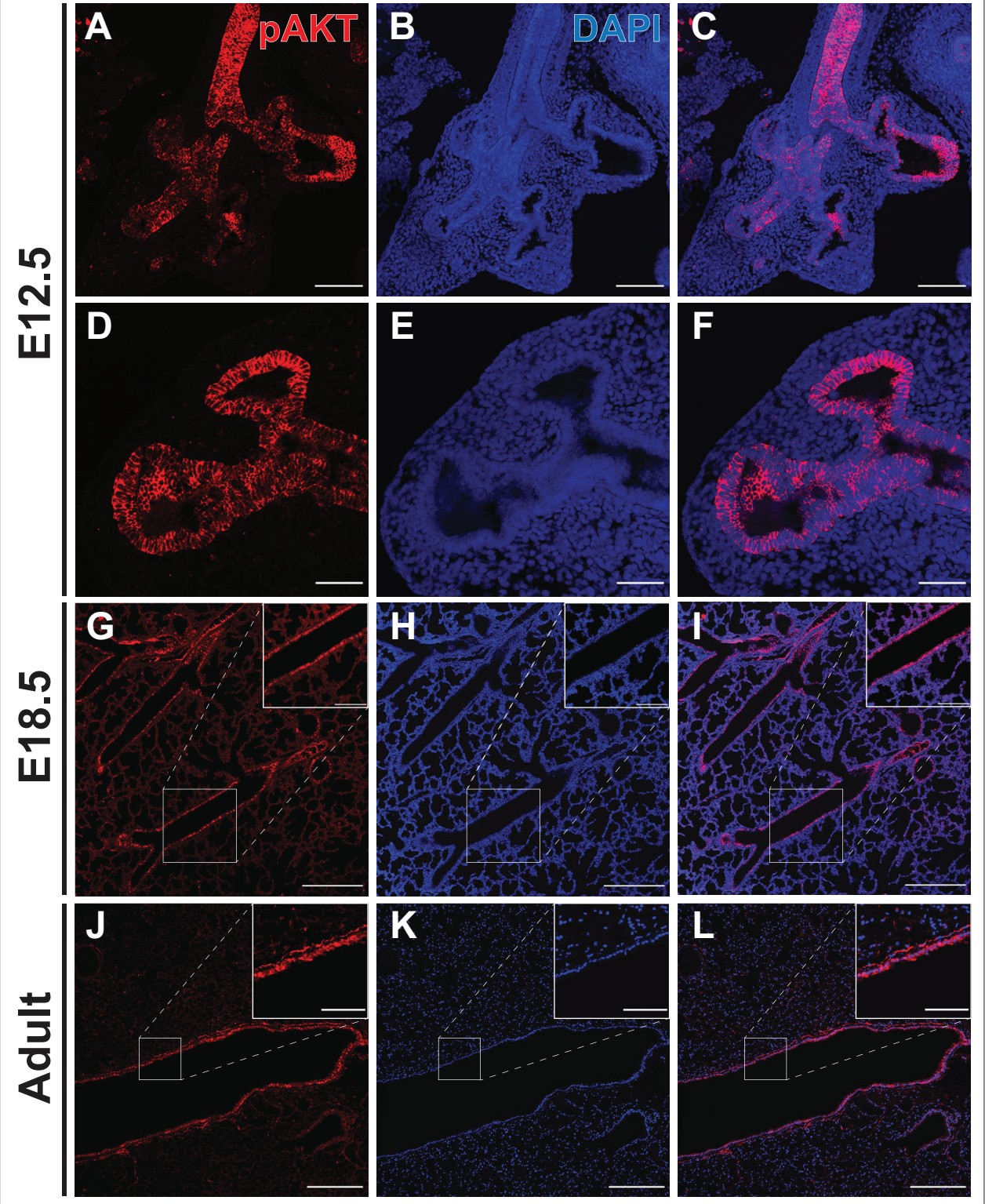

**Figure 5.** Phospho-AKT staining in the developing lung epithelium. (**A–F**) Strong staining for AKT phosphorylated at Ser473 (pAKT) was observed in the early lung epithelium at E12.5, with minimal staining seen in the mesenchyme. Staining intensity was highest in the proximal lung, but appeared to be pan-epithelial with relatively intense staining also observed at the distal tips. (**G–I**) By E18.5, highest levels of pAKT staining were detected in epithelial cells lining the large conducting airways. (**J–L**) Strong pAKT staining was observed in the conducting airway epithelium, with notable expression in the sub-epithelial airway mesenchyme (see inset). Scale bars: 100 μm (**A–C**), 50 μm (**D–F**), 250 μm (**G–L**), 100 μm (inset G-I), 50 μm (inset J-L).

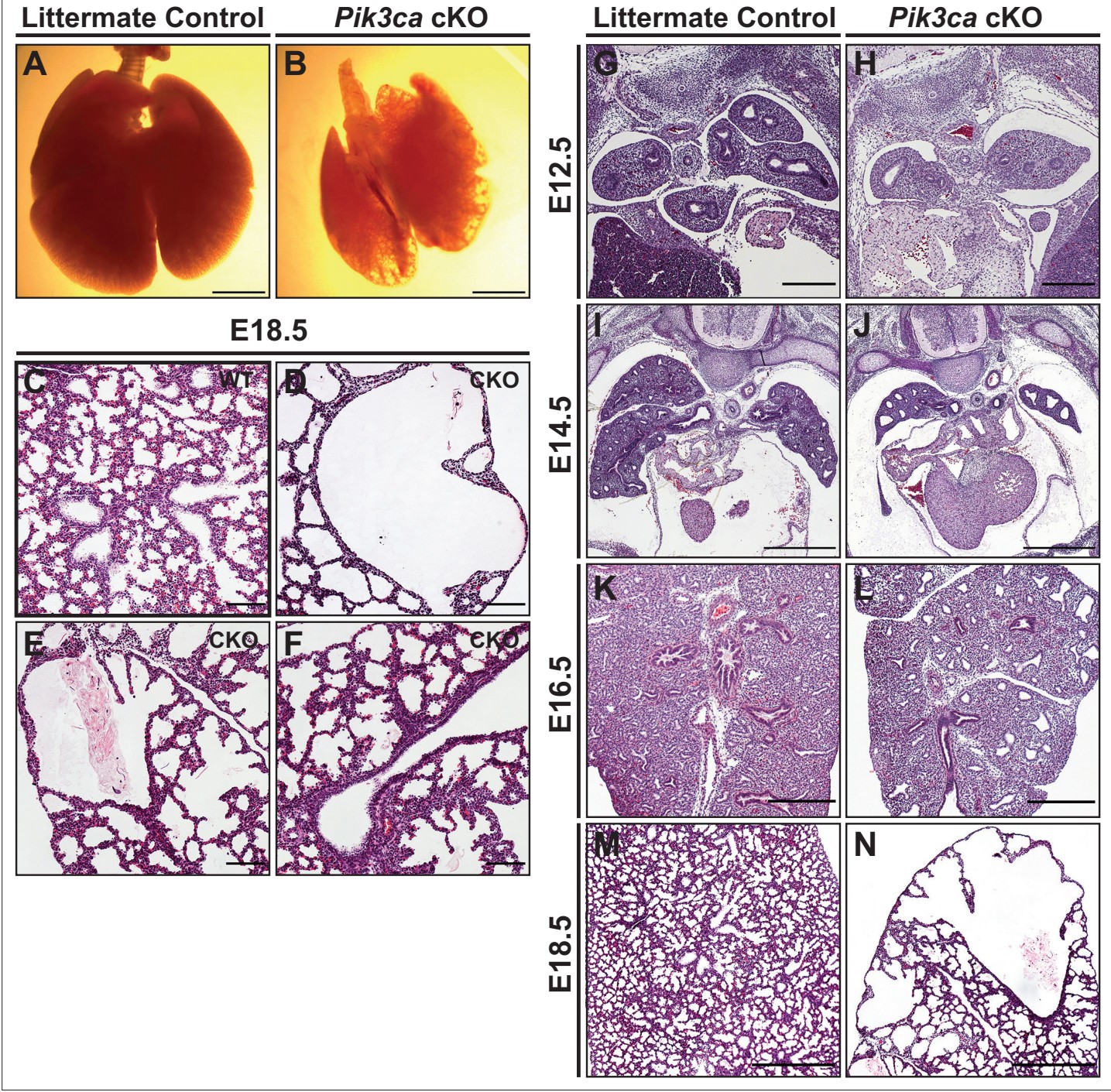

**Figure 6.** Conditional deletion of *Pik3ca* from the developing lung epithelium results in impaired branching morphogenesis and cystic pulmonary hypoplasia. (**A–B**) Whole-mount images show cystic areas throughout the lungs of *Pik3ca cKO* embryos at E18.5. The lungs are smaller in size compared to littermate controls. (**C–F**) Widefield imaging of H&E stains shows numerous cystic areas in *Pik3ca cKO* and a paucity of conducting airways at E18.5. The majority of the lung tissue consists of simplified alveolar structures. (**G–N**) A time-series of *Pik3ca cKO* embryonic lungs show a decreased number of epithelial branches and smaller lung size at E12.5 and E14.5 (**G–J**). Dilated airspaces are evident by E16.5 (**K–L**), and large cystic structures with simplified alveoli are seen at E18.5 (**M–N**). Scale bars: 2.5 mm (**A–B**), 100 µm (**C–H**), 250 µm (**I–L**), 500 µm (**M–N**).

The online version of this article includes the following figure supplement(s) for figure 6:

**Figure supplement 1.** Conditional deletion of Pik3ca with Shh-Cre results in loss of epithelial pAKT staining.

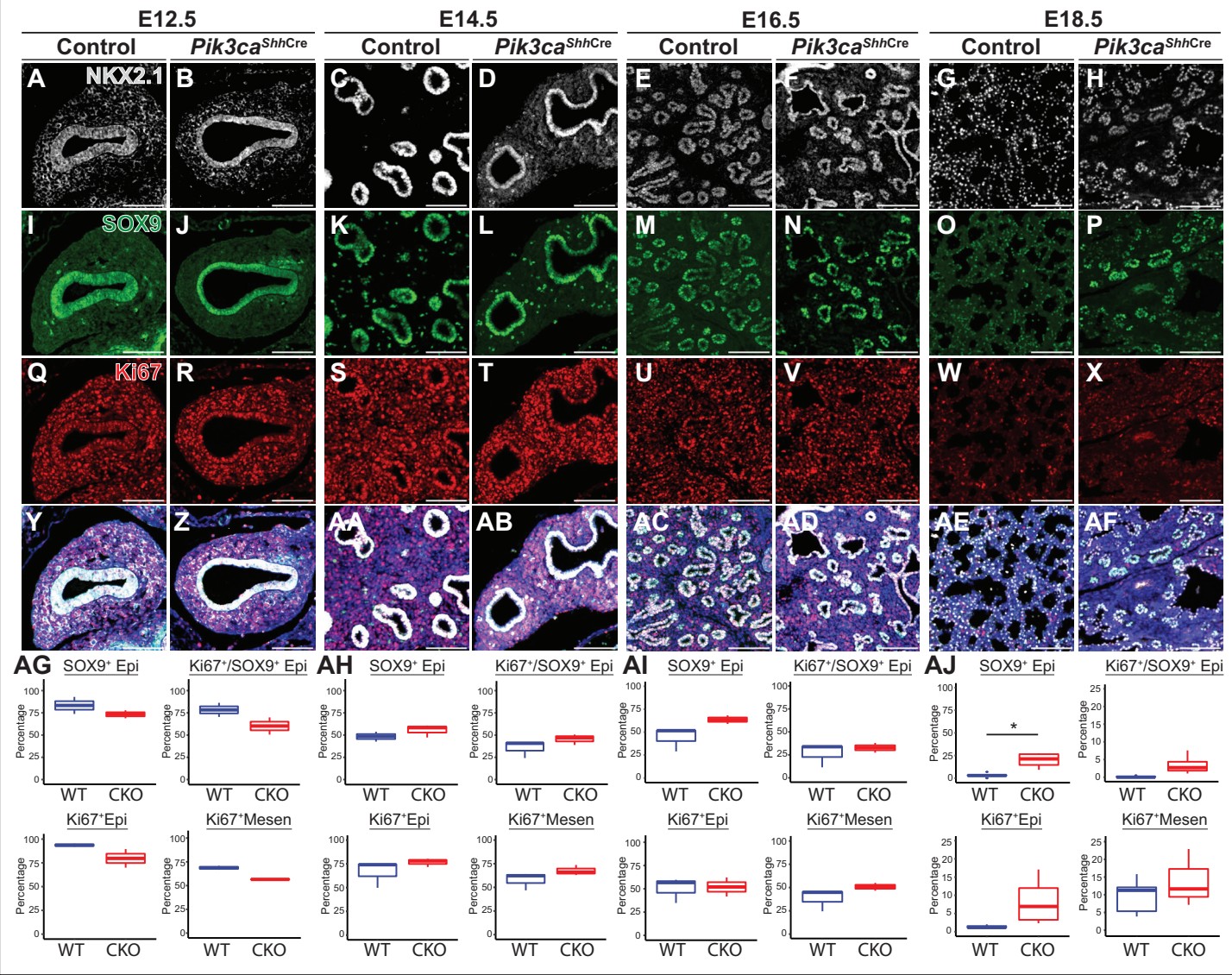

**Figure 7.** Loss of Pik3ca in the developing lung epithelium leads to persistence of the SOX9+epithelial progenitor cell population at E18.5. (**A–AF**) The relative numbers of NKX2−1+epithelial cells, SOX9+ epithelial progenitor cells, and fraction of proliferating (Ki67+) cells within each population were assessed and quantified using immunofluorescence microscopy from E12.5 to E18.5. (AG-AI) No significant differences were observed in the relative total numbers of SOX9+ EPCs or proliferating SOX9+ EPCs from E12.5 to E16.5. (AJ) At E18.5, significant numbers of SOX9+ EPCs were still present, in contrast to littermate controls. This difference did not appear to be accounted for by a change in proliferation. (n=2–3 embryos per genotype at each timepoint) Scale bars: 100 µm. * p<0.05. Comparisons were not statistically significant, unless noted otherwise.

*Foxj1* mRNA levels. (Figure Y-MM, NN-OO) In contrast, the impact on the number of SOX2+ airway epithelial cells was much less dramatic. From E12.5-E16.5, there appears to be a decrease in the number of airway branches consistent with an impairment in braching morphogenesis, also evident in H&E staining shown in *Figure 6*. However, the airways present appear to be lined with epithelial cells expressing SOX2+ at an intensity comparable to littermate controls. At E18.5, *Sox2* mRNA expression levels were overall decreased by an average of 30% (but did not reach statistical significance; *Figure 8A–X*, NN). Taken together, these data suggest that there is a significant impairment of airway epithelial differentiation following epithelial-specific loss of PI3K signaling, with a more subtle impact on the total number of SOX2 +airway epithelial cells overall.

Taken together, these data demonstrate that epithelial-specific PI3K signaling is required for proper proximal/distal patterning of the lung during development. Both in vivo and in vitro data demonstrate that loss of PI3K signaling causes persistence of the SOX9+ lung epithelial progenitors

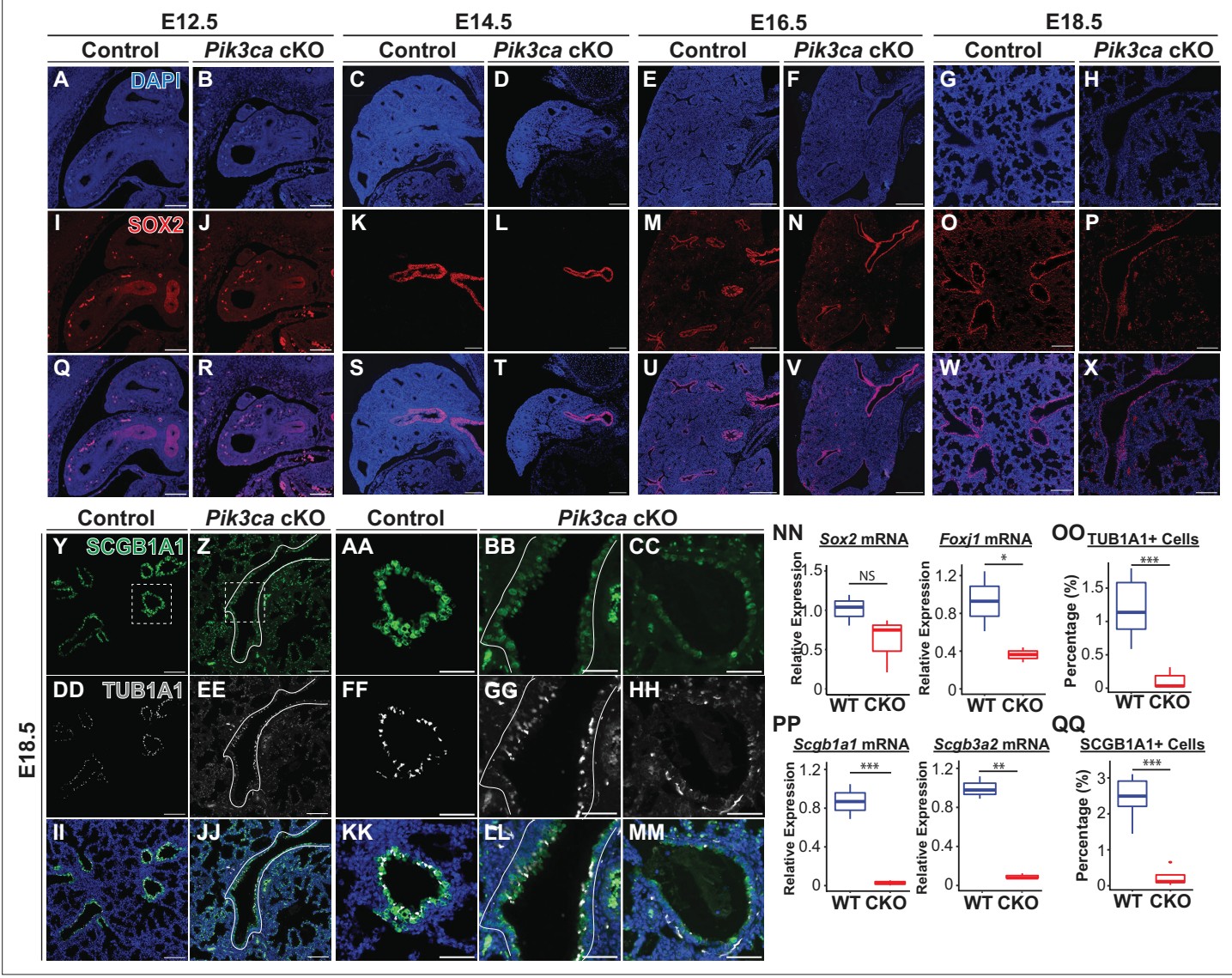

**Figure 8.** Loss of Pik3ca in the developing lung epithelium leads to impaired airway epithelial cell differentiation. (**A–X**) SOX2 expression is observed in the airway epithelium from E12.5 to E18.5. (**NN**) Although the total number of SOX2 + cells present at E18.5 appears to be decreased in images A-X, the relative expression of Sox2 mRNA is not significantly changed. (**Y–MM, NN–PP**) A marked decrease in both the total number of SCGB1A1+secretory cells and TUB1A1+ciliated cells, and the mRNA expression levels of related transcripts was observed at E18.5. Scale bars: 100 µm (A-X, except E16.5 timepoint; **Y–Z, DD–EE, II–JJ**), 250 µm (**E–F, M–N, U–V**), 50 µm (AA-CC, FF-HH, KK-MM). N=5 *Pik3ca cKO* lungs and N=6 wild-type lungs were included for all cell counts based on immunofluorescence microscopy (OO, QQ) N=3 biological replicates per genotype for qPCR (NN, PP). Two-tailed Student's t-test used for NN-QQ. * p<0.05, ** p<0.01, *** p<0.001.

into late embryonic life, inhibiting normal proximal lung epithelial differentiation and resulting in mispatterning of both the airway and alveolar structures of the lung. Finally, this requirement for PI3K signaling demonstrates that careful comparative analysis using both chromatin accessibility and gene expression data provides higher sensitivity to identify novel regulators of development that may be unrecognized from expression data alone.

## Discussion

The mature mammalian lung is composed of dozens of specialized cell types that are necessary to permit gas exchange and facilitate terrestrial life (*Morrisey and Hogan, 2010*; *Swarr and Morrisey, 2015*; *Basil et al., 2020*; *Basil and Morrisey, 2020*). While many of the major transcription factors

and signaling pathways that drive differentiation of lung cells during development have been defined, mechanisms by which these signals are integrated into the chromatin state to establish and maintain cellular identity remains incompletely understood. Here, we have generated a map of the chromatin accessibility landscape during SOX9+ lung epithelial progenitor cell development, which we have made readily available to the scientific community through the LGEA Web Portal (https://research. cchmc.org/pbge/lunggens/mainportal.html), a product of the NIH LungMap Project (*Du et al., 2015*; *Ardini-Poleske et al., 2017*; *Du et al., 2017*). Accessible chromatin regions identified in our study are highly correlated with histone post-translational modificiations normally associated with active, functional cis-regulatory elements, and changes in the accessibility of these regions of chromain are associated with changes in developmental gene expression. Changes in chromatin accessibility were preferentially located at putative enhancers, highlighting the importance of targeted chromatin remodeling at select enhancers in the development of the SOX9+ lung epithelial progenitors.

Our combinatorial approach to the analysis of both mRNA and chromatin accessibility data led to the observation that PI3K was a key regulator of EPC development. Accessible regions of chromation, particularly those regions that became more accessible as development proceeded, were significantly enriched nearest genes associated associated with PI3K signaling. Signaling through the PI3K pathway, as measured by pAKT immunostaining, was highest in the proximal lung epithelium throughout development. In vivo, a paucity of airways were observed in the *Pik3ca cKO* lungs; although the airways present were lined by SOX2+ epithelium, mature secretory and ciliated cell markers were decreased. *Pik3ca cKO* mice also demonstrated extensive distal cysts and simplified alveoli. Both results emphasize the need for an intact gradient of PI3K signaling during development for proper differentiation of the proximal and distal portions of the lung. Follow-up studies using inducible Cre drivers to inactivate the PI3K pathway in more specific cell populations at specific stages of development will be needed to understand the time- and context-dependency of PI3K signaling in lung development.

The markedly abnormal lung epithelium seen in *Pik3ca cKO* animals imply a critical role for PI3K in lung development, raising several future questions. The role of PI3K signaling in directing branching morphogenesis remains incompletely defined. Although conditional deletion of *Pik3ca* in the developing lung epithelium in vivo appears to lead to impaired branching morphogenesis, lung explant models employing pharmacologic inhibitors of PI3K signaling, and a recent study that conditionally deleted *Pik3ca* in the lung mesenchyme during mouse lung development show an increase in branching morphogenesis. (*Carter et al., 2014*; *Dai et al., 2022*) The overlapping and unique roles of PI3K signaling in the epithelium and mesenchyme warrant further investigation. Second, the receptor/ ligand pairs modulating PI3K signaling in the developing lung remain to be defined. FGF signaling, particularly FGF7 and FGF10 signaling through FGFR1/2 has been shown to play a key role in initial outgrowth of the lung bud and subsequent branching morphogenesis. However, abrogation of FGF7 or FGF10 signaling through FGFR1/2 causes apoptosis and failure of lung epithelial outgrowth, not expansion of SOX9+ progenitor cells as observed presently, and thus the phenotypes observed in *Pik3ca cKO* animals cannot simply be explained by a downstream effector of the FGF7/10-FGFR1/2 signaling axis; EGF, other FGFs, and IGF are all potential regulators to be explored. It is worth noting that the 'bimodal' pattern of pAKT staining early in development may be a result of FGF10-FGFR2 signaling at the branch tips, with signaling through additional receptor-ligand pairs acting to pattern and direct differentiation of the proximal epithelium.

Third, our data suggest that the PI3K pathway may regulate chromatin state during lung development. Studies from cancer, where aberrant activation of PI3K signaling is common, have shown that pAKT can directly interact with a number of key chromatin regulatory (CR) complexes to modulate chromatin state (*Yang et al., 2019*). We speculate that the PI3K signaling pathway serves as the central 'switch-board' integrating signals received from diverse receptor ligand-pairs during lung epithelial development, and in turn directly modulates chromatin state, including chromatin accessibility, through interactions between pAKT and key CR complexes to direct lung epithelial differentiation (*Figure 9*). Future studies investigating the interface of PI3K signaling within the lung epithelium on chromatin state, cellular differentiation, and cell-type identity will be of significant interest in defining these mechanisms.

Finally, our data highlight the dearth of data regarding the molecular mechanisms by which chromatin accessibility changes are regulated during differentiation of the SOX9+ EPC and other lung progenitor populations. It remains unknown which of the dozens of proteins and complexes that

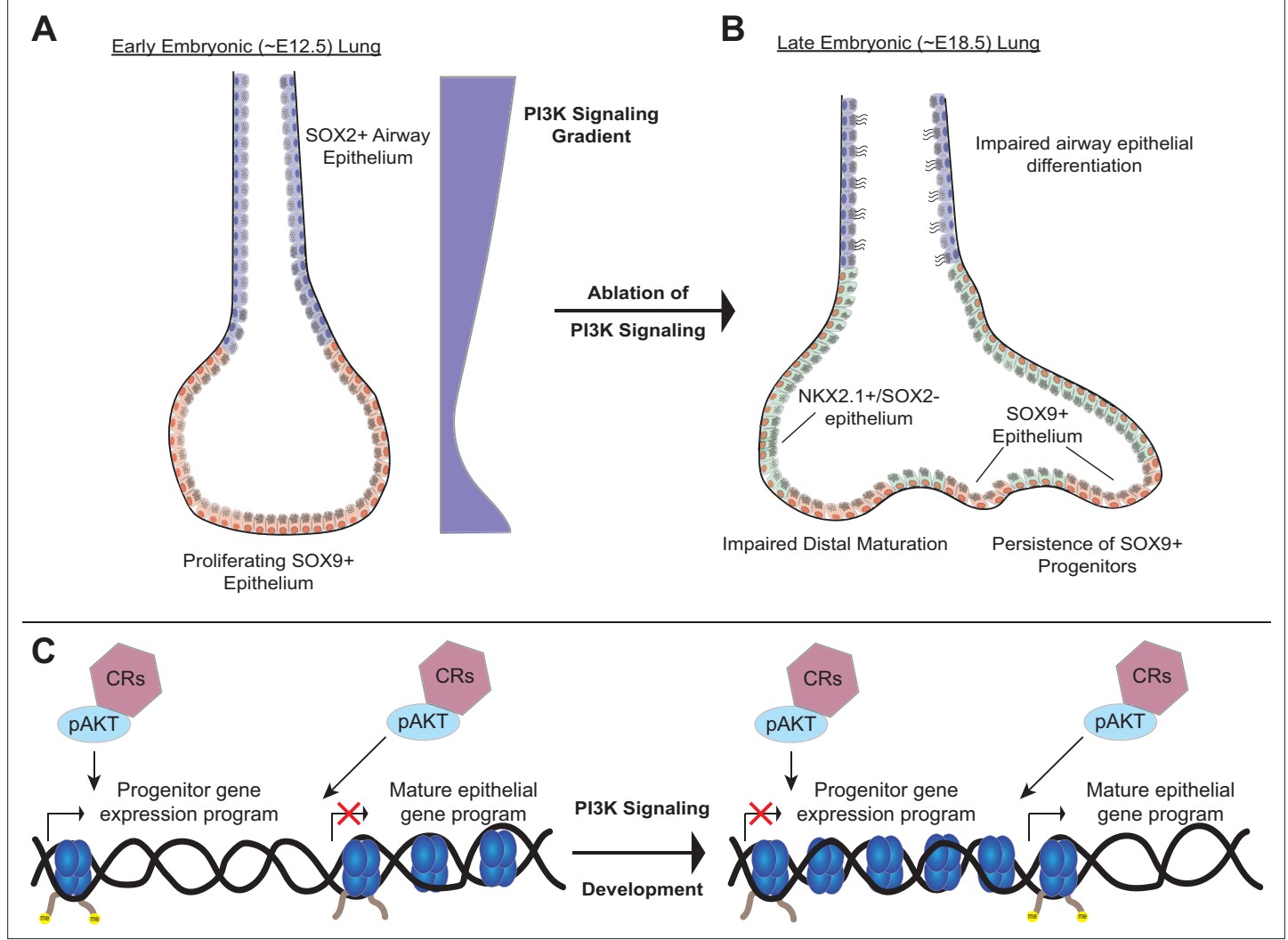

**Figure 9.** A model of the role of PI3K signaling in the developing lung epithelium. (**A**) During normal lung epithelial development, a proximal-to-distal gradient of PI3K signaling patterns the lung epithelium, with highest levels in the developing conducting airways and distal tips (early in development). (**B**) Genetic ablation of PI3K signaling causes persistence of the SOX9+progenitor cells, and impairs epithelial differentiation of alveolar and conducting airway epithelial cells. (**C**) PI3K signaling may directly modulate chromatin accessibility to promote differentiation and pattern cell-type identity during lung epithelial development.

are known to remodel chromatin are specifically required for development and differentiation of the lung epithelium. Studies in other tissues imply that specific CR complexes interface with major developmental transcription factors to mediate chromatin accessibility, and that chromatin accessibility changes are an integral part of cell-type identity. Given the promising progress of epigenomic therapies in cancer (*Bates, 2020*), there is an urgent need for in vivo genetic studies targeting specific CR complexes and CRISPR-based epigenome engineering strategies (*Holtzman and Gersbach, 2018*) to define the epigenomic mechanisms of lung homeostasis and disease.

# Materials and methods
## Animal husbandry

Experiments were performed on a mixed C57BL/6, CD-1 background. All animal work was performed under the approval and guidance of the Cincinnati Children's Hospital Institutional Animal Care and Use Committee (IACUC). The Tg(Sox9-EGFP)[EB209Gsat] mouse line (MGI ID: 3844824) was used for isolation of SOX9+epithelial cells, using the methods described below (*Gong et al., 2017*). The *Shh*[tm1((EGFP/]

cre)Cjt) (JAX stock # 055622) and *Pik3ca*tm1Jjz (JAX stock #017704) lines were used to conditionally delete *Pik3ca* from the developing lung epithelium (*Harfe et al., 2004*; *Zhao et al., 2006*).

## Single-cell RNA sequencing analysis

Publicly available scRNA-seq data from EpCAM sorted E11 mouse lung epithelium were combined with publicly available E16 whole lung scRNA using Seurat's *FindIntegrationAnchors*, using only genes found in both datasets (*Kuwahara et al., 2020*). After integration, SeuratWrappers and Monocle 3 were used to create a linage trajectory after indicating E11 epithelial cells as the root of the model, minimal_branch_len = 18. After sub-setting out all epithelial cells, gene modules were calculated using Monocle3's *find_gene_modules*, resolution = 1e-3, with *graph_test* results as input, q_value <0.01. Unless indicated all default settings for Mononcle3 were used. Marker genes for E16 distal and proximal epithelial cells were generated by comparing only epithelial clusters using Seurat's *FindMarkers*. Genes in gene modules of interest were analyzed by Toppfunn's functional enrichment analyses to determine biological process related to specific modules.

## SOX9+ lung epithelial progenitor cell isolation

Timed matings were performed by placing a Tg(Sox9-EGFP)EB209Gsat male reporter mouse with a wild-type female, and the presence of a copulation plug the following morning was used to define gestational day (GD) 0.5. Lungs were microdissected from embryos isolated from pregnant females at E11.5 and E16.5 (n=3 at each timepoint). Lung lobes were isolated from trachea, eosphagus and surrounding tissue, minced with fine scissors, and then were digested with 1 mL 10 X TrypLE (Gibco) for 5–10 min at 37 °C with agitation. Cells were filtered (70 μm and 35 μm filters, respectively) and washed with FACS buffer (HBSS buffer(Gibco), 2% FBS, 25 mM HEPES buffer pH 7.0, 2 mM EDTA pH 7.4) twice. Next, cells were labeled with anti-CD326 (eBioscience #17-5791-82), anti-CD31 (eBioscience #25-0311-82) and anti-CD45 (eBioscience #15-0451-82) at a 1:200 dilution for 10 min at 4 C. Cells were then washed again, and treated with 7-AAD (BioLegend) (5 μL per 100 μL FACS buffer). SOX9+ lung epithelial cells were isolated by sorting for GFP+/CD326+/CD31-/CD45-/7-AAD- using a BD/FACSAria II cell sorter. Cells were sorted directly into Trizol (ThermoFisher Scientific) for RNA isolation, or enriched cell-culture media (HBSS buffer (Gibco), 25 mM HEPES buffer, 50% FBS) for ATAC-Seq library preparation.

## RNA-Seq library preparation, sequencing, and bioinformatic analysis

Isolated SOX9+ lung epithelial cells sorted directly into Trizol were sent to GeneWiz, LLC for RNA isolation and library preparation. After Trizol/chloroform extraction, RNA isolation was completed with the RNeasy Micro Plus kit per manufacturer's instructions (Qiagen). SMART-Seq v4 Ultra-Low Input RNA kits (TaKaRa Bio) were used to generated polyA selected libraries from total RNA. Sequencing was performed on an Illumina HiSeq platform, using 2x150 bp configuration. Quality trimming and adapter clipping of sequencing reads was performed with Trimmmatic (*Bolger et al., 2014*). Read quality was assessed before and after trimming using the FastQC program. Reads were then aligned against the mouse reference genome (mm10) using the STAR aligner (*Dobin et al., 2013*). Duplicate reads were flagged and removed using the MarkDuplicates program from Picard tools. Per-gene read counts for GENECODE release M12 (GRCm38.p5) were computed using the Rsubread R package, with duplicate reads removed (*Frankish et al., 2019*; *Liao et al., 2019*). Gene counts represented as counts per million (CPM) were first nominalized using the TMMmethod in the edgeR R package, and genes with 25% of samples with a CPM <1 were removed. The data were transformed using the VOOM function from the limma R package (*Law et al., 2014*). Differential gene expression was performed using a linear model with the limma package. Heat maps and principal component analysis plots were generated in R. GO enrichment analysis for the biological process was performed using the R package GAGE (*Luo et al., 2009*). Expression of select gene targets were displayed using the boxplot function of ggplot2() in R with tpm counts generated by Stringtie (*Pertea et al., 2015*; *Wickham, 2016*).

## ATAC-Seq library preparation, sequencing, and bioinformatic analysis

Isolated SOX9+lung epithelial cells were sorted into HBSS buffer (Gibco) containing 50% FBS and 25 mM HEPES buffer (Gibco). Nuclei were isolated, the transposase reaction was performed with

Nextera Tn5 transposase (Illumina), and PCR amplification was performed according to previously published protocols (*Buenrostro et al., 2015a*). Sequencing was performed by GeneWiz, LLC on an Illumina HiSeq platform, using 2x150 bp configuration. Reads were quality filtered and adapters were trimmed using Trimmomatic, version 0.39 (*Bolger et al., 2014*). Reads were aligned to mouse genome mm10 using bowtie2 (*Langmead and Salzberg, 2012*). Duplicated reads were removed with picard tools, and blacklisted regions were filtered out using bedtools (*Amemiya et al., 2019*). Peak calling was performed with MACS2 (version 2.2.1.20160309) using the additional parameters '—nomodel', '—shift –100', and '—extsize 200'. The MACS2 'bdgdiff' function was used to identify differentially accessible regions. A q-value (FDR) of 0.05 was used to determine significance for both analyses. Peak annotation, gene ontology analysis of nearest genes, and quantification of ENCODE ChIP-Seq read density by ATAC-seq peak location analyses were performed using Homer and heatmaps were generated using the aheatmap() module in the R software package (*Heinz et al., 2010*). The R software package was used to measure and display relationships between ATAC-Seq peak location and nearest gene neighbor expression values. ChIP-Seq data for H3K4me1, H3K4me3, and H3K27ac were accessed either directly from the ENCODE Project website (https://www.encodeproject.org/) or via the UCSC genome browser (https://genome.ucsc.edu/) (*ENCODE Project Consortium, 2012*; *Sloan et al., 2016*; *Davis et al., 2018*). A list of ENCODE accession numbers and generating laboratories for these datasets appears in *Supplementary file 7*. Genomic interval plots were generated using the UCSC genome browser with the GRCm38/mm10 genome builds. Paired expression and chromatin accessibility modeling was performed using the methods and software described previously (*Duren et al., 2017*). Stringtie was used to generate tpm based expression matrices as input into PECA from merged bam files for all replicates at each developmental timepoint (*Pertea et al., 2015*). Replicated ATAC-Seq peaks were used as the chromatin accessibility input to PECA.

## Histology and immunofluorescence microscopy

Tissues were fixed in 4% paraformaldehyde, embedded in paraffin wax, and sectioned at 5 μm intervals. Slides were stained with hematoxylin using standard protocols, as previously described (*Herriges et al., 2014*; *Swarr et al., 2019*). Immunofluorescence staining was performed using the following antibodies: anti-NKX2.1 (guinea pig, Seven Hills, 1:500), anti-SOX9 (rabbit, Millipore, 1:100), anti-KI67 (mouse, BD Pharmigen, 1:100), anti-SCGB1A1 (rabbit, Seven Hills, 1:500), anti-TUB1A1 (mouse, Sigma Aldrich, 1:1000), and anti-SOX2 (mouse, Santa Cruz, 1:100). Immunofluorescence for phosphorylated-ATK (pAKT) was performed with anti-pAKT (rabbit, Cell Signaling, 1:100), followed by multi-step detection with the Biotin-Tyramide signal amplification kit (Perkin Elmer) with a 15-min exposure time. TUNEL staining was performed using the TACS-XL Blue Label kit, per manufacturer's instructions (Trevigen). All slides were mounted with Prolong Gold Antifade medium (Invitrogen), and were then imaged on either a Nikon Eclipse 90i widefield, or Nikon A1R GaAsP Inverted Confocal Microscope. Cell type quanitification based on immunofluorescence microscopy images was performed using the Nikon Elements Advanced Analysis software suite.

## Whole-mount imaging

Tracheal lung tissue isolated at E12.5, E13.5, and E14.5 was subject to whole mount immunofluorescence staining as described (*Sinner et al., 2019*). Embryonic tissue was fixed in 4% PFA overnight and then stored in 100% MeOH at –20 °C. Whole-mounts were permeabilized in Dent's Bleach (4:1:1 MeOH: DMSO: 30%$H_2O_2$) for 2 hr, and were then taken from 100% MeOH to 100% PBS. Following washes, tissue was blocked in a 5% (w/v) blocking solution for 2 hr, and was then incubated overnight at 4 °C in primary antibody solution. After washes in PBS, whole-mounts were incubated with a secondary antibody overnight at 4 °C. Samples were then washed, dehydrated and cleared in Murray's Clear. Images of whole-mounts were obtained using confocal microscopy (Nikon A1R). Imaris imaging software was used to convert z-stack image slices obtained using confocal microscopy to 3D renderings of whole-mount samples.

## Quantitative real-time PCR

A single-cell suspension was generated, using the methods described above, from either lung explant cultures or embryonic mouse lungs isolated at E18.5 from Pik3ca cKO embryos and littermate controls. Epithelial cells were isolated using sheep anti-rat Dynabeads (Thermo Fisher) with rat anti-mouse

CD326 antibody (Thermo Fisher). Total RNA was subsequently isolated from the bead-sorted epithelial cells using Trizol (Invitrogen), per the manufacturer's protocol. cDNA was synthesized from total RNA by using the SuperScript IV strand synthesis system (Invitrogen). Quantitative real-time PCR was performed using the SYBR Green system (Applied Biosystems) with primers listed in *Supplementary file 8*. *GAPDH* and *Tbp* expression values were used to control for RNA quantity. Two-tailed Student's T-Test was used for all comparisons involving two groups.

## Data access

The data sets from this study have been submitted to the NCBI Gene Expression Omnibus (GEO; https://www.ncbi.nlm.nih.gov/geo/) under accession number: GSE188239.

## Acknowledgements

We acknowledge the assistance of the Research Flow Cytometry Core and Confocal Imaging Core at Cincinnati Children's Hospital Medical Center. We are thankful for the generous grant support from the following sources: National Institutes of Health grants 7K08HL130666, 5R01HL156860 (DTS) and Parker B Francis Fellowship Award (DTS). DK, SF, MG, SJ, SSJ, and DS all performed experiments related to this manuscript. DK, SF, JS, MG, and DS performed the data analysis in this manuscript. DK, SF, WZ, and DS all contributed to writing the manuscript.

## Additional information

### Funding

| Funder | Grant reference number | Author |
|---|---|---|
| National Institutes of Health | K08HL130666 | Daniel T Swarr |
| National Institutes of Health | K08HL140178 | William Zacharias |
| National Institutes of Health | R01HL144774 | Debora Sinner |
| Cincinnati Children's Hospital Medical Center | Proctor Scholar Award | Daniel T Swarr |
| National Institutes of Health | 5R01HL156860 | Daniel T Swarr |

The funders had no role in study design, data collection and interpretation, or the decision to submit the work for publication.

### Author contributions

Divya Khattar, Daniel T Swarr, Conceptualization, Data curation, Formal analysis, Supervision, Funding acquisition, Validation, Investigation, Methodology, Writing – original draft, Project administration, Writing – review and editing; Sharlene Fernandes, Data curation, Formal analysis, Validation, Investigation, Methodology, Writing – original draft, Writing – review and editing; John Snowball, Minzhe Guo, Data curation, Formal analysis, Visualization, Writing – review and editing; Matthew C Gillen, Data curation, Formal analysis, Investigation, Visualization, Writing – review and editing; Suchi Singh Jain, Validation, Investigation, Visualization, Methodology, Writing – review and editing; Debora Sinner, Formal analysis, Investigation, Visualization, Methodology, Writing – original draft, Writing – review and editing; William Zacharias, Conceptualization, Data curation, Formal analysis, Supervision, Funding acquisition, Validation, Investigation, Visualization, Methodology, Writing – original draft, Project administration, Writing – review and editing

### Author ORCIDs

Debora Sinner http://orcid.org/0000-0002-0704-5223
William Zacharias http://orcid.org/0000-0002-2643-0610
Daniel T Swarr http://orcid.org/0000-0002-7305-0442

### Ethics

This study was performed in strict accordance with the recommendations in the Guide for the Care and Use of Laboratory Animals of the National Institutes of Health. All of the animals were handled according to approved institutional animal care and use committee (IACUC) protocol (#2019-016) of Cincinnati Children's Hospital Medical Center. The protocol was approved by the Cincinnati Children's Animal Care and Use Committee (Animal Welfare Assurance # A3108-01).

### Decision letter and Author response

Decision letter https://doi.org/10.7554/eLife.67954.sa1
Author response https://doi.org/10.7554/eLife.67954.sa2

## Additional files

### Supplementary files

• Supplementary file 1. Genes identified to be differentially expressed between E11.5 and E16.5 in SOX9+lung epithelial cells sorted using flow cytometry are listed.

• Supplementary file 2. Significantly accessible regions identified using MACS2 in SOX9+lung epithelial cells sorted using flow cytometry at E11.5 are listed.

• Supplementary file 3. Significantly accessible regions identified using MACS2 in SOX9+lung epithelial cells sorted using flow cytometry at E16.5 are listed.

• Supplementary file 4. Regions of differentially accessible chromatin in SOX9+lung epithelial cells sorted using flow cytometry, with increased accessibility at E11.5 compared to E16.5, are listed.

• Supplementary file 5. Regions of differentially accessible chromatin in SOX9+lung epithelial cells sorted using flow cytometry, with increased accessibility at E16.5 compared to E11.5, are listed.

• Supplementary file 6. Regions of accessible chromatin in SOX9+lung epithelial cells sorted using flow cytometry, that did not undergo statistically significant changes in accessibility between E11.5 and E16.5 (common regions), are listed.

• Supplementary file 7. Information for ENCODE datasets used in this study are listed.

• Supplementary file 8. Oligonuclotide DNA sequences used as primers for RT-PCR (qPCR) in this study are listed.

• Transparent reporting form

### Data availability

Sequencing data have been deposited in the GEO database, under the accession code GSE188239, GSE188230, and GSE188237.

The following datasets were generated:

| Author(s) | Year | Dataset title | Dataset URL | Database and Identifier |
| --- | --- | --- | --- | --- |
| Khattar D, Fernandes S, Snowball J, Guo M, Gillen MC, Sinner D, Zacharias W, Swarr DT | 2022 | Sox9+ Progenitor Cell RNA-Seq | http://www.ncbi.nlm.nih.gov/geo/query/acc.cgi?acc=GSE188230 | NCBI Gene Expression Omnibus, GSE188230 |
| Khattar D, Fernandes S, Snowball J, Guo M, Gillen MC, Sinner D, Zacharias W, Swarr DT | 2022 | Sox9+ Progenitor Cell ATAC-Seq | http://www.ncbi.nlm.nih.gov/geo/query/acc.cgi?acc=GSE188237 | NCBI Gene Expression Omnibus, GSE188237 |
| Khattar D, Fernandes S, Snowball J, Guo M, Gillen MC, Sinner D, Zacharias W, Swarr DT | 2022 | PI3K signaling specifies proximal-distal fate by driving a developmental gene regulatory network in SOX9+ mouse lung progenitors | http://www.ncbi.nlm.nih.gov/geo/query/acc.cgi?acc=GSE188239 | NCBI Gene Expression Omnibus, GSE188239 |

The following previously published datasets were used:

| Author(s) | Year | Dataset title | Dataset URL | Database and Identifier |
|---|---|---|---|---|
| Rin B | 2014 | H3K4me1 ChIP-seq on embryonic 16.5 day mouse lung | https://www.encodeproject.org/experiments/ENCSR387YSD/ | ENCODE, ENCSR387YSD |
| Rin B | 2014 | H3K4me3 ChIP-seq on embryonic 16.5 day mouse lung | https://www.encodeproject.org/experiments/ENCSR295PFM/ | ENCODE, ENCSR295PFM |
| Rin B | 2014 | H3K27ac ChIP-seq on embryonic 16.5 day mouse lung | https://www.encodeproject.org/experiments/ENCSR140UEX/ | ENCODE, ENCSR140UEX |
| Kuwahara A, Lewis AE, Coombes C, Leung FS, Percharde M, Bush JO | 2020 | Delineating the early transcriptional specification of the mammalian trachea and esophagus | https://doi.org/10.7272/Q6WW7FVB | Dryad Digital Repository, 10.7272/dryad.Q6WW7FVB |
| Whitsett JA, Xu Y | 2015 | E16.5 mouse lung, scRNA_Seq | https://research.cchmc.org/pbge/lunggens/celltype_E16_p3.html | LungGens, E16.5 scRNA-Seq |

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
