## [Editor Report]

This study reveals the importance of PI3K in regulating early aspects of lung development including the establishment of a proximal-distal gradient in cell fate in the lung endoderm. The data presented will provide a rich resource for further examination of the role of PI3K and the transcription factor *Sox9* in lung endoderm development.

---

## [Decision Letter]

**Decision letter after peer review:**

Thank you for submitting your article "PI3K signaling specifies proximal-distal fate by driving a distinct gene regulatory network in *SOX9*+ lung progenitors" for consideration by *eLife*. Your article has been reviewed by 3 peer reviewers, and the evaluation has been overseen by Edward Morrisey as the Senior Editor. The reviewers have opted to remain anonymous.

The reviewers have discussed their reviews with one another, and the Senior Editor has drafted this to help you prepare a revised submission.

Essential revisions:

1) Temper conclusions as to the effects of the small molecular inhibitors on epithelial versus mesenchymal cell types.

2) Perform the additional time points and immunostaining as described by the reviewers.

*Reviewer #1 (Recommendations for the authors):*

This is an interesting article starting with the analysis of paired gene expression and chromatin accessibility data during lung development and concluding with a proposed role for Pi3K in branching morphogenesis and proximal distal differentiation.

The in vivo data look very interesting.

Some issues with the current experiments however make them hard to interpret.

1) while the experiments in Figure 6 show an increase in branching morphogenesis after treatment with different inhibitors, it is unclear whether this is because inhibition of Pi3K in the epithelium or mesenchyme.

2) Similarly it is difficult to assess whether the effect on *Sox9* epithelial progenitors is due to the inhibitor acting on the epithelium or mesenchyme.

3) The authors could try to separate the epithelium from the mesenchyme and treat their explants with the inhibitor again.

4) The authors should show increased branching and *Sox9* in E13.5 Shhcre-Pik3ca lungs.

5) whole mount staining as performed in Figure 1 for *Sox2* and *Sox9* would be much more illustrative of changes in proximal distal differentiation.

6) Lastly in the abstract the authors mention that prior to E13.5, *SOX9*+ progenitors are multipotent, generating both airway and alveolar epithelium, but are selective alveolar progenitors later in development. To further investigate this the authors isolated *Sox9* positive progenitors at 11.5 and 16.5. The authors then as expected find some genes being differentially expressed in the progenitors at these different time points. However, while these changes in expression likely reflect the narrowed differentiation potential of the *Sox9*+ EPCs at E16.5; does it really help to explain how *Sox9*+EPCs at E11.5 differentiate into proximal epithelium?

qPCR in Figure 8 reflects the lack of airways but doesn’t reflect their differentiation, it appears differentiation in club and ciliated cells still occurs but appears delayed. Differentiation of the bronchial epithelium occurs after *Sox9*+ EPCs have differentiated into *sox2*^+^ airway cells.

It is unclear if the differentiation of the *sox2*^+^ airway epithelium is delayed or whether Pik3ca plays a role in the differentiation of these *sox2*^+^ airway epithelial cells.

*Reviewer #2 (Recommendations for the authors):*

In the manuscript by Khattar et al., they have taken a novel combinatorial approach in lung biology using epigenetic and transcriptomic data to identify active transcriptional networks during lung development. Their PECA analysis uncovered PI3K signaling as an important pathway in lung development. As noted, PI3K signaling is not entirely novel for lung development, but the current study has provided some additional understanding.

Confirmation of some of the regulated components of PI3K signaling in the PECA analysis should be confirmed by qPCR, iHC, or RNAscope/in situ. In particular, those components that such as Grhl2, Hbp1, or Sox4.

Phospho-AKT is the only read out used to support PI3K activity. AKT is regulated by diverse signaling mechanisms, and there are multiple kinase signaling pathways downstream of PI3K. Any additional read out (see above comment) would provide more support for your conclusion.

Likewise, the current imaging that reports a PI3K signaling gradient is not convincing. At E12.5, pAKT appears expressed in all the epithelium. Denoting proximal vs distal epithelium with the use of *Sox2* and *Sox9* on multiple development time points will improve this conclusion. IN addition, the authors should consider assessment of pAKT in specific cell types in later development.

The mechanism for expansion of *SOX9*+ EPCs is unclear. There is decreased proliferation. As mentioned in the public review, could this be at the expense of loss the proximal *sox2*^+^ EPCs. Again, *SOX2*-*SOX9* co-staining at several time points during branching morphogenesis would help support this. Alternatively, is Grhl2 involved?

In addition, branching abnormalities have been implicated in alveolar specification defects and could explain your differentiation anomalies.

Similar to above, deletion of PI3Kac should be confirmed by gene expression and pAKT and/or other read outs of PI3K signaling.

*Reviewer #3 (Recommendations for the authors):*

The following critiques are aimed at further strengthening the findings and conclusions.

1. Much of the predictions that led to identification of PI3k was not followed up, and these are missed opportunities. For example, the relationship shown in Figure 3 of PI3K link to transcription factors was not investigated. RNAseq of the mutant will be informative to reveal possible links.

2. Based on histological data in Fig7 and 8, there seem to be opposing phenotypes in the in vitro experiment with increase branch number compared to in vivo phenotype with simplified and dilated alveolar region, which typically is a result of decreased branching. The discrepancy needs to be addressed, perhaps by assaying the mutants at earlier stages to determine if there is increased branching.

3. The analysis of the in vivo phenotype is superficial. For example, the persistent of *Sox9* and Ki67 cell proliferation are not quantified, and the data are not of high resolution. It is not clear what is driving reduced club and ciliated cell phenotype, given that *Sox2* is not altered. Deeper analysis to tie together the ATACseq and RNAseq analysis as mentioned in point 1 will be important to put all findings into one cohesive study.

4. The increase in branching phenotype is unique and robust. What is the cellular mechanism behind this? Is there a change in branching signals such as Fgf10, etc?

5. Given the nature of PI3K signaling, much of the regulation in this pathway occurs via protein-level regulation. It will be useful to have a closer examination of the transcriptional regulations of both the factors that are known to be regulated post-transcriptionally, and compared to factors in the rest of the pathway.

6. Pik3ca histology is reminiscent of CPAM. Is there a connection?

---

## [Author Response]

Reviewer #1 (Recommendations for the authors):This is an interesting article starting with the analysis of paired gene expression and chromatin accessibility data during lung development and concluding with a proposed role for Pi3K in branching morphogenesis and proximal distal differentiation.The in vivo data look very interesting.Some issues with the current experiments however make them hard to interpret.1) while the experiments in Figure 6 show an increase in branching morphogenesis after treatment with different inhibitors, it is unclear whether this is because inhibition of Pi3K in the epithelium or mesenchyme.2) Similarly it is difficult to assess whether the effect on Sox9 epithelial progenitors is due to the inhibitor acting on the epithelium or mesenchyme.3) The authors could try to separate the epithelium from the mesenchyme and treat their explants with the inhibitor again.

See response to comment #1 (public review). We have removed the inhibitor experiments and focused on the interpretation and analysis of the more robust in vivo knockout model in this revision.

4) The authors should show increased branching and Sox9 in E13.5 Shhcre-Pik3ca lungs.

We have added time series data from E12.5 through E18.5 to the manuscript for Pik3ca^ShhCre^ embryos.

These in vivo data suggest that there is an impairment of branching morphogenesis with epithelial specific loss of PI3K signaling during development, contrary to the results of our previously presented in vitro explant data. We have therefore focused on the interpretation of the in vivo phenotype to clarify presentation.

Recently published data from Xing Y, et al., (Frontiers in Cell and Developmental Biology, 2022) demonstrate that aberrant activation of PI3K signaling within the lung mesenchyme increases branching morphogenesis and impairs airway epithelial differentiation. It is possible that the increase in branching morphogenesis seen in our in vitro data is due to mesenchymal PI3K signaling, partial pathway inhibition (compared to complete blockade in the genetic mouse model), or timing of pathway inhibition (starting at E12.5 in the in vitro model, with earlier pathway inactivation in the mouse model). These questions will be best addressed with conditional deletion and pathway activation in both the epithelium and mesenchyme in vivo, which is beyond the scope of the current manuscript, and the suggestions from the reviewer will be useful in thinking about those experiments.

5) whole mount staining as performed in Figure 1 for Sox2 and Sox9 would be much more illustrative of changes in proximal distal differentiation.

The protocol used for whole mount imaging in Figure 1 is most robust at earlier developmental times. Our quantification data from in vivo Pik3ca knockout shows the majority of phenotypes appreciated in this model present late, between E16-18.5, limiting the utility of the whole mount assay to define these phenotypes in our hands. We have therefore provided both multiple images and quantifications of all 2d stains from the later stages by traditional sectioning to maximize interpretability.

6) Lastly in the abstract the authors mention that prior to E13.5, SOX9+ progenitors are multipotent, generating both airway and alveolar epithelium, but are selective alveolar progenitors later in development. To further investigate this the authors isolated Sox9 positive progenitors at 11.5 and 16.5. The authors then as expected find some genes being differentially expressed in the progenitors at these different time points. However, while these changes in expression likely reflect the narrowed differentiation potential of the Sox9+ EPCs at E16.5; does it really help to explain how Sox9+EPCs at E11.5 differentiate into proximal epithelium?qPCR in Figure 8 reflects the lack of airways but doesn’t reflect their differentiation, it appears differentiation in club and ciliated cells still occurs but appears delayed. Differentiation of the bronchial epithelium occurs after Sox9+ EPCs have differentiated into sox2^+^ airway cells.It is unclear if the differentiation of the sox2^+^ airway epithelium is delayed or whether Pik3ca plays a role in the differentiation of these sox2^+^ airway epithelial cells.

See response to Question #4 (Public Review).

Reviewer #2 (Recommendations for the authors):In the manuscript by Khattar et al., they have taken a novel combinatorial approach in lung biology using epigenetic and transcriptomic data to identify active transcriptional networks during lung development. Their PECA analysis uncovered PI3K signaling as an important pathway in lung development. As noted, PI3K signaling is not entirely novel for lung development, but the current study has provided some additional understanding.Confirmation of some of the regulated components of PI3K signaling in the PECA analysis should be confirmed by qPCR, iHC, or RNAscope/in situ. In particular, those components that such as Grhl2, Hbp1, or Sox4.Phospho-AKT is the only read out used to support PI3K activity. AKT is regulated by diverse signaling mechanisms, and there are multiple kinase signaling pathways downstream of PI3K. Any additional read out (see above comment) would provide more support for your conclusion.Likewise, the current imaging that reports a PI3K signaling gradient is not convincing. At E12.5, pAKT appears expressed in all the epithelium. Denoting proximal vs distal epithelium with the use of Sox2 and Sox9 on multiple development time points will improve this conclusion. IN addition, the authors should consider assessment of pAKT in specific cell types in later development.

We have added coronal sections of phospho-AKT (pAKT) immunofluorescence microscopy of the early lung (E12.5) to better illustrate the distribution of PI3K signaling activity. The strongest signal is seen in the proximal airway epithelium (developing trachea), but signal is present throughout the lung epithelium and there does appear to be a second focus of increased signal intensity at the distal epithelial tips. By E18.5 pAKT signal intensity appears to be far greater within the airway epithelium than any other location in the lung. Because pAKT signal appears to be present in all of the airway epithelial cells at E18.5 and adult lungs, we did not feel there was any additional benefit to performing co-stains with additional airway epithelial markers, particularly given that pAKT staining alone is technically challenging.

With regard to additional readouts of PECA-identified targets, we performed QPCR on epithelial cells identified by bead sorting for EPCAM from E18.5 embryos and saw minimal difference in expression in PIK3ca KO vs control lungs. These results are shown in Author response image 1.

**Author response image 1. sa2fig1:** 

We interpret these differences in QPCR to be not surprising, as the PECA model integrates both chromatin accessibility data and target gene expression to identify core regulators within a specific cellular context. Extensive prior literature has shown that transcription factor binding and other secondary regulation controls TF activity based on differential cellular state, and thus differences in the regulators themselves do not validate or invalidate the overall network given that the predicted targets. The major finding of PECA was to direct us toward PI3K signaling as an interesting regulator of proximal/distal specification, and our knockout experiments clearly show this is the case. We have adjusted the discussion of PECA in the manuscript to reflect these caveats.Finally, as the reviewer points out, there are multiple differential readouts of the upstream signaling mechanisms that drive PI3K signaling in the lung. The extensive crosstalk of these pathways is complex and beyond the scope of the current work. We focused specifically on downstream effects of PI3K signaling, and agree with the reviewer that future genetic experiments to precisely dissect the upstream signals driving these effects in different developmental compartments will be important.

The mechanism for expansion of SOX9+ EPCs is unclear. There is decreased proliferation. As mentioned in the public review, could this be at the expense of loss the proximal sox2^+^ EPCs. Again, SOX2-SOX9 co-staining at several time points during branching morphogenesis would help support this. Alternatively, is Grhl2 involved?

We have added a time series from E12.5 through E18.5 of *SOX2* and *SOX9* immunofluorescence microscopy, with quantification of the percentage of *SOX9*+ epithelium and proliferating *SOX9*+ cells at each timepoint. in vivo, there does not appear to be a significant difference in the percentage of *SOX9*+ epithelial cells from E12.5-E16.5. At E18.5, there are very few remaining *SOX9*+ epithelial cells in wild-type lungs, but there appears to be persistence of *SOX9*+ lung epithelial cells in the distal alveolar regions. These data suggest that the most likely explanation is not proliferative expansion of the *SOX9*+ progenitor cell population, but a persistence of the *SOX9*+ progenitor cell state and impaired differentiation.

In addition, branching abnormalities have been implicated in alveolar specification defects and could explain your differentiation anomalies.

We think the primary defect in the lungs is in airway epithelial differentiation, but agree that future work (beyond the scope of this paper) should also further investigate the impact on alveolar epithelial cell differentiation in this mouse model.

Similar to above, deletion of PI3Kac should be confirmed by gene expression and pAKT and/or other read outs of PI3K signaling.

We have added pAKT immunofluorescence microscopy of Pik3ca^ShhCre^ lung and matched littermate controls, which demonstrate complete loss of pAKT signal from the lung epithelium of Pik3ca^ShhCre^ mice, confirming successful pan-class I PI3K pathway inhibition.

Reviewer #3 (Recommendations for the authors):The following critiques are aimed at further strengthening the findings and conclusions.1. Much of the predictions that led to identification of PI3k was not followed up, and these are missed opportunities. For example, the relationship shown in Figure 3 of PI3K link to transcription factors was not investigated. RNAseq of the mutant will be informative to reveal possible links.

We performed QPCR on epithelial cells identified by bead sorting for EPCAM from E18.5 embryos and saw minimal difference in expression in PIK3ca KO vs control lungs. These results are shown in Author response image 1.

We interpret these differences in QPCR to be not surprising, as the PECA model integrates both chromatin accessibility data and target gene expression to identify core regulators within a specific cellular context. Extensive prior literature has shown that transcription factor binding, and other secondary regulation controls TF activity based on differential cellular state, and thus differences in the regulators themselves do not validate or invalidate the overall network given that the predicted targets. The major finding of PECA was to direct us toward PI3K signaling as an interesting regulator of proximal/distal specification, and our knockout experiments clearly show this is the case. We have adjusted the discussion of PECA in the manuscript to reflect these caveats.

However, we do think this is a very important area of focus for future investigation, and we hope that ongoing transcriptomic and chromatin accessibility studies with this conditional knockout mouse line (which we think is beyond the scope for the current manuscript) will help better address these questions.

2. Based on histological data in Fig7 and 8, there seem to be opposing phenotypes in the in vitro experiment with increase branch number compared to in vivo phenotype with simplified and dilated alveolar region, which typically is a result of decreased branching. The discrepancy needs to be addressed, perhaps by assaying the mutants at earlier stages to determine if there is increased branching.

We have added a time series from E12.5 to E18.5 for our core imaging data sets. We agree that there appears to be a decrease in branching morphogenesis in Pik3ca^ShhCre^ lungs, which contrasts with the increase in branching morphogenesis observed in the in vitro explant models. One potential explanation is that the increase in branching morphogenesis observed in the in vitro explant model is due to inhibition of PI3K signaling in the mesenchyme, since pharmacologic treatment obviously results in pathway inhibition in both compartments of the lung. Recently published data from Xing Y, et al., (Frontiers in Cell and Developmental Biology, 2022) demonstrate that aberrant activation of PI3K signaling within the lung mesenchyme increases branching morphogenesis and impairs airway epithelial differentiation. Other possible explanations about the discrepancy between the in vitro explant data and Pik3ca^ShhCre^ mouse model would be timing (earlier pathway inhibition in in vivo model) or degree (partial versus complete in vitro versus in vivo models) of pathway inhibition. We agree that these are fascinating questions, but will be best addressed by additional in vivo studies using Cre-lox technology to both ablate as well as constitutively activate the pathway in the epithelium and mesenchyme to further delineate cell-type specific effects of PI3K signaling.

3. The analysis of the in vivo phenotype is superficial. For example, the persistent of Sox9 and Ki67 cell proliferation are not quantified, and the data are not of high resolution. It is not clear what is driving reduced club and ciliated cell phenotype, given that Sox2 is not altered. Deeper analysis to tie together the ATACseq and RNAseq analysis as mentioned in point 1 will be important to put all findings into one cohesive study.

We have added a developmental time series, from E12.5 to E18.5 and have quantified cell types within these figures that we hope better illustrates the phenotype. From E12 to E16.5 there are no significant differences in the number of proliferating epithelium (or proliferating *Sox9*+ epithelium) or *Sox9*+ epithelium. At E18.5, there are significantly more *Sox9*+ lung epithelial cells in the Pik3ca^ShhCre^ lungs, which is probably best described as a persistence of this progenitor cell population (rather than proliferative expansion of *Sox9*+ epithelium). Immunofluorescence microscopy of *Sox2* from E12.5 to E18.5 again shows fewer airway branches (as discussed above), but the airway epithelium otherwise appears to express *Sox2* in a normal fashion. At E18.5, the mean expression level of *Sox2* mRNA in Pik3ca^ShhCre^ lungs is modestly decreased but does not reach statistical significance. In contrast, there is a marked reduction in the secretoglobin proteins Scgb1a1 and Scgb3a2 at both the mRNA and protein level, and the ciliated cell marker Foxj1 (at mRNA level). Although we cannot exclude that there is a smaller but significant impairment in the generation of *sox2*^+^ epithelium, it appears that the more significant phenotype present is the differentiation of airway epithelium into mature airway epithelium. We anticipate that our follow-up studies will refine the precise molecular mechanisms by which PI3K signaling directs differentiation of the lung epithelium.

4. The increase in branching phenotype is unique and robust. What is the cellular mechanism behind this? Is there a change in branching signals such as Fgf10, etc?

As discussed above, the apparent conflict between the in vitro explant model (increased branching) and in vivo epithelial-specific pathway ablation (decreased branching) is interesting and raises important questions about the precise roles that PI3K signaling plays in directing branching morphogenesis. Recently published data from Xing Y, et al., (Frontiers in Cell and Developmental Biology, 2022) demonstrate that aberrant activation of PI3K signaling within the lung mesenchyme increases branching morphogenesis and impairs airway epithelial differentiation. It is possible that the increase in branching morphogenesis seen in our in vitro data is due to mesenchymal PI3K signaling, partial pathway inhibition (compared to complete blockade in the genetic mouse model), or timing of pathway inhibition (starting at E12.5 in the in vitro model, with earlier pathway inactivation in the mouse model). These questions will be best addressed with conditional deletion and pathway activation in both the epithelium and mesenchyme in vivo, which is beyond the scope of the current manuscript, and the suggestions from the reviewer will be useful in thinking about those experiments.

5. Given the nature of PI3K signaling, much of the regulation in this pathway occurs via protein-level regulation. It will be useful to have a closer examination of the transcriptional regulations of both the factors that are known to be regulated post-transcriptionally, and compared to factors in the rest of the pathway.

We agree that both the inputs into the PI3K signaling pathway (e.g. FGFs, EGF, IGFs) during lung development and transcriptional output of the pathway are important open questions. While FGF10FGFR2 signaling at the distal tip during early branching morphogenesis has been well studied, and may explain the increased pAKT staining we see at the branch tips (in addition to the high level of pAKT in the proximal epithelium/airway epithelium), we feel that there is important unexplored biology in how the PI3K pathway integrates these diverse signaling pathways during development, and potentially modulates chromatin state to effect transcriptional change. We hope that ongoing transcriptomic and chromatin accessibility studies with this conditional knockout mouse line, and well as in vivo models employing constitutive pathway activation (which we think is beyond the scope of the current manuscript) will help better address these questions.

6. Pik3ca histology is reminiscent of CPAM. Is there a connection?

Very interesting question. Our previous work on congenital pulmonary airway malformations implicated PI3K signaling as a pathway possibly dysregulated in CPAMs. There is an evolution of cystic lesions relatively late in lung development in the Pik3ca^ShhCre^ lungs, but the other most notable aspect of the phenotype observed in these animals is an impairment in airway epithelial differentiation, with a marked decrease in ciliated and secretory cells markers. In contrast, there is a significant increase in the number of airway epithelial cells with increased expression levels of ciliated and secretory cell markers in CPAM lesions. Somatic mosaic mutations in PIK3CA are seen in a number of pediatric disorders (referred to as the Pik3ca-related overgrowth spectrum (PROS)). It is conceivable that activating somatic mutations in Pik3ca (or other PI3K pathway components) within the developing lung could underly the pathogenesis of CPAM, but this warrants further investigation in human samples (whole-exome sequencing) and a constitutive pathway activation mouse model (which will complement the pathway inactivation model presented in this manuscript).